# Wnt11 acts on dermomyotome cells to guide epaxial myotome morphogenesis

**Ann Kathrin Heilig[1,2,3], Ryohei Nakamura[1], Atsuko Shimada[1], Yuka Hashimoto[1], Yuta Nakamura[1], Joachim Wittbrodt[2], Hiroyuki Takeda[1]\*, Toru Kawanishi[1]\***

[1]Department of Biological Sciences, Graduate School of Science, University of Tokyo, Tokyo, Japan; [2]Centre for Organismal Studies, Heidelberg University, Heidelberg, Germany; [3]Heidelberg Biosciences International Graduate School, Heidelberg, Germany

**Abstract** The dorsal axial muscles, or epaxial muscles, are a fundamental structure covering the spinal cord and vertebrae, as well as mobilizing the vertebrate trunk. To date, mechanisms underlying the morphogenetic process shaping the epaxial myotome are largely unknown. To address this, we used the medaka *zic1/zic4*-enhancer mutant *Double anal fin* (*Da*), which exhibits ventralized dorsal trunk structures resulting in impaired epaxial myotome morphology and incomplete coverage over the neural tube. In wild type, dorsal dermomyotome (DM) cells reduce their proliferative activity after somitogenesis. Subsequently, a subset of DM cells, which does not differentiate into the myotome population, begins to form unique large protrusions extending dorsally to guide the epaxial myotome dorsally. In *Da*, by contrast, DM cells maintain the high proliferative activity and mainly form small protrusions. By combining RNA- and ChIP-sequencing analyses, we revealed direct targets of Zic1, which are specifically expressed in dorsal somites and involved in various aspects of development, such as cell migration, extracellular matrix organization, and cell-cell communication. Among these, we identified *wnt11* as a crucial factor regulating both cell proliferation and protrusive activity of DM cells. We propose that dorsal extension of the epaxial myotome is guided by a non-myogenic subpopulation of DM cells and that *wnt11* empowers the DM cells to drive the coverage of the neural tube by the epaxial myotome.

**\*For correspondence:**
htakeda@bs.s.u-tokyo.ac.jp (HT);
toru.kawanishi@bs.s.u-tokyo.ac.jp (TK)

**Competing interest:** The authors declare that no competing interests exist.

## Editor's evaluation

This study addresses an interesting and underexplored question in developmental biology, specifically cell migration and muscle development. It builds upon prior analysis of the medaka *Double anal fin* (*Da*) mutants by using detailed bioinformatic and time-lapse analysis to explain dorsal somite extension. The authors show that dorsal muscle morphogenesis is actively guided by dorsal dermomyotome cells, rather than being passively shaped by physical constraints alone. Looking downstream of *Da*, they show that Wnt signaling is central to dorsal extension of the epaxial myotome and propose that similar functions may shape the dorsal musculature across vertebrates.

## Introduction

Active locomotion, which is powered by skeletal muscles in vertebrates, is critical for animals to survive. Vertebrate skeletal muscles consist of axial muscles (head, trunk, and tail muscles) and appendicular muscles (limb muscles). Axial muscles first arose in the chordates to stabilize and enable side-to-side movement of the body axis. In jawed vertebrates, the subdivision of axial muscles into epaxial (dorsal) and hypaxial (ventral) muscles led to an increased range of movement: dorsoventral undulation in fish and lateral movements in terrestrial vertebrates (*Goodrich, 2004*; *Romer and Parsons, 1986*; *Fetcho,*

*1987*; *Sefton and Kardon, 2019*). Among these muscles, epaxial muscles are characterized by their unique anatomical structure, which extends dorsally and surrounds the vertebrae. This morphology also ensures mechanical support and protection of the vertebrae and the spinal cord inside. While we have a detailed understanding of how the myotome, precursors of epaxial and hypaxial muscles, differentiates from a somitic compartment called the dermomyotome (DM) (*Kalcheim et al., 1999*; *Gros et al., 2009*; *Hollway and Currie, 2005*), we only begin to understand the cellular and molecular mechanisms of the subsequent morphogenetic processes generating the epaxial muscles. Previous studies in rats and mice suggested that myocytes of the epaxial myotome do not actively migrate dorsally but are guided by external forces (*Deries et al., 2010*; *Deries et al., 2012*). However, what exerts such forces to drive extension of epaxial myotome is still unclear.

Fish have been extensively utilized to study myotome development thanks to the transparency of their bodies throughout embryonic development (*Wolff et al., 2003*; *Stellabotte et al., 2007*; *Nguyen et al., 2017*; *Ganassi, 2018*). Additionally, their epaxial trunk muscles have a simple structure consisting of only one anatomical unit (reviewed in *Sefton and Kardon, 2019*). Like other vertebrates, fish myotomes, on either side of the neural tube, extend dorsally after somite differentiation and eventually cover the neural tube by the end of embryonic development (*Figure 1A*). The spontaneous medaka (*Oryzias latipes*) mutant *Double anal fin* (*Da*) displays a particular epaxial myotome morphology, in which the dorsal ends of the left and right epaxial myotome fail to extend sufficiently and thus do not cover the neural tube at the end of embryonic development. Previous studies demonstrated that the dorsal trunk region of the *Da* mutant is transformed into the ventral one, including not only the myotome but also the body shape, skeletal elements, pigmentation, and fin morphology (*Figure 1B and D*; *Ishikawa, 1990*; *Ohtsuka et al., 2004*). Given the unique morphological features, the medaka *Da* mutant is an excellent model to study the morphogenesis of epaxial myotome. Genetic analysis of the *Da* mutant revealed that this phenotype is due to a dramatic reduction of the expression of the transcription factors *zic1* and *zic4* in the dorsal somites (*Figure 1C and E*), and identified *zic1*/*zic4* as master regulators of trunk dorsalization (*Ohtsuka et al., 2004*; *Kawanishi et al., 2013*). The down-regulation of *zic1/zic4* specifically in the dorsal somites is caused by the insertion of a large transposon, disrupting the dorsal somite enhancer of *zic1/zic4* (*Moriyama et al., 2012*; *Inoue et al., 2017*). While the function and downstream targets of Zic1 and Zic4 have been studied in nervous system development and somitogenesis (*Li, 2006*; *Pan et al., 2011*; *Himeda et al., 2013*; *Hong and Saint-Jeannet, 2017*; *Aruga and Millen, 2018*), the molecular mechanism of how these Zic genes control dorsal trunk morphogenesis has not been investigated so far.

Here, we describe the morphogenetic process of the formation of the epaxial myotome of the back, which we termed 'dorsal somite extension' (*Figure 1A*). By in vivo time-lapse imaging we uncovered its cellular dynamics; during dorsal somite extension, dorsal DM cells reduce their proliferative activity and subsequently form unique large protrusions extending dorsally, guiding the epaxial myotome towards the top of the neural tube. We also found that these DM cells form a subpopulation that gives rise to non-myotomal cell lineages during embryonic development. In the *Da* mutant, by contrast, DM cells keep their high proliferative activity and mainly form small protrusions. Mechanistically, we identified a Zic1 downstream-target gene, *wnt11* (former *wnt11r Postlethwait et al., 2019*), as a crucial factor for dorsal somite extension, and demonstrated that Wnt11 regulates cellular behavior of dorsal DM cells by promoting protrusion formation and negatively regulating proliferation. We thus propose an unprecedented process of epaxial myotome morphogenesis driven by a non-myogenic population of DM cells during embryogenesis.

## Results

Our previous study showed that *zic1* and *zic4* expression starts at embryonic stages and persists throughout life (*Kawanishi et al., 2013*). Phenotypic analysis of homozygous adult *Da* mutants implies long-term participation of Zic-downstream genes in the formation of dorsal musculatures, which eventually affects the external appearance of the fish adult trunk. Here, we examined the initial phase of this long-term dorsalization process. In the following of the study, we will focus on *zic1*, since *zic1* and *zic4* are expressed in an identical fashion with overlapping functions in trunk dorsalization of medaka, and *zic4* is expressed slightly weaker than *zic1* (*Moriyama et al., 2012*; *Kawanishi et al., 2013*).

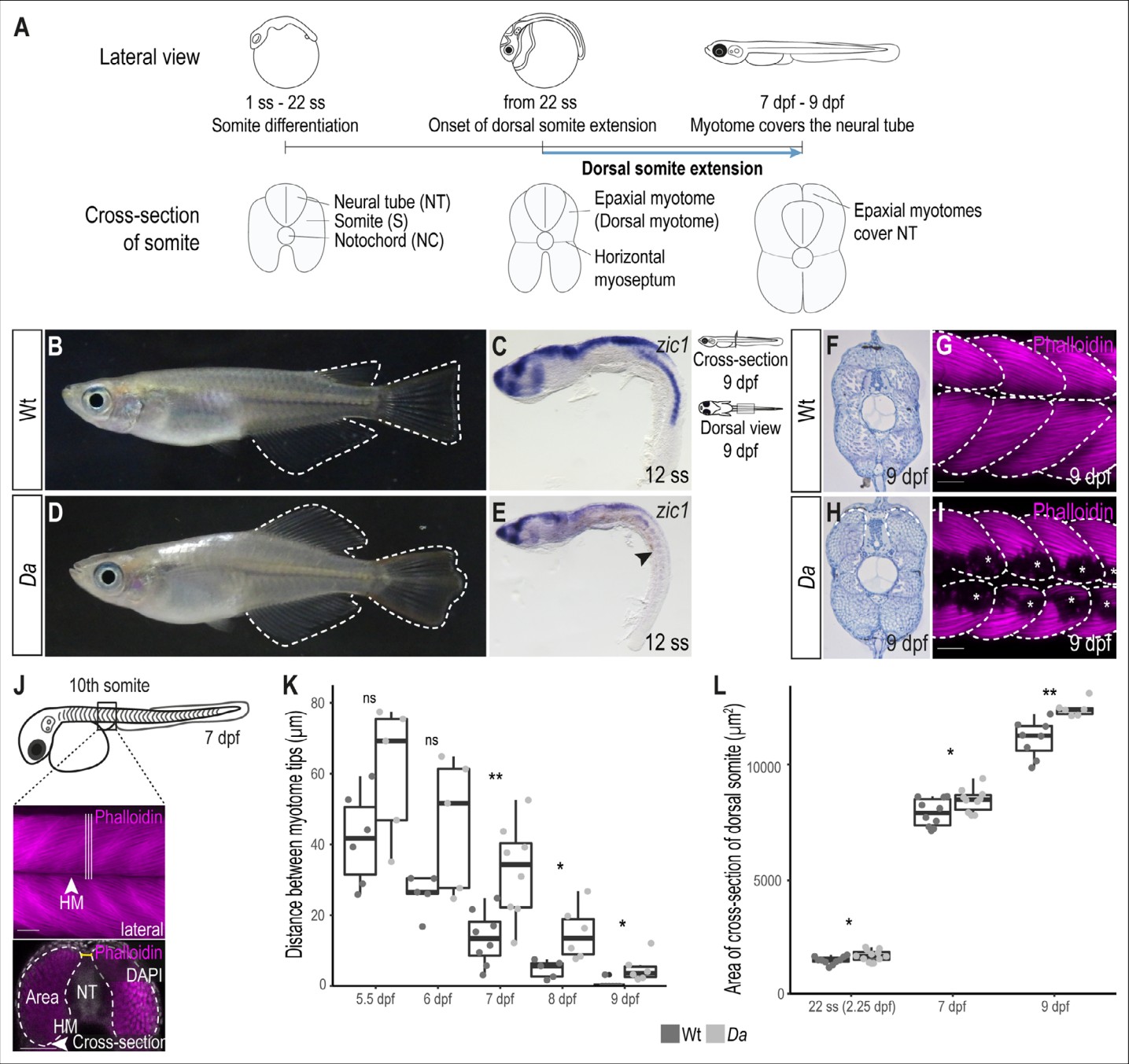

**Figure 1.** The epaxial myotome of the *Da* mutant fails to cover the neural tube at the end of embryonic development. (**A**) Schematic representation of dorsal somite extension which results in the full coverage of the neural tube by the epaxial myotomes at the end of embryonic development. (**B**) Lateral view of adult Wt medaka. Dorsal, caudal, and anal fins are outlined. (**C**) Lateral view of whole-mount in situ hybridization against *zic1* in a 12 ss (1.7 dpf, stage 23) Wt embryo. *zic1* expression can be observed in the brain, neural tissues and the dorsal somites. (**D**) Lateral view of adult *Da* mutant. Dorsal, anal and caudal fins are outlined. The dorsal trunk region resembles the ventral trunk region. (**E**) Lateral view of whole-mount in situ hybridization against *zic1* of a 12 ss *Da* embryo. *zic1* expression can be observed in the brain and the neural tissues, but is drastically decreased in the dorsal somites (arrowhead). (**F, H**) Cross-sections of tail regions of hematoxylin stained 9 dpf embryos. Dorsal ends of myotomes are outlined. In Wt, the left and the right myotome come in close contact at the top of the neural tube and form a gapless muscle layer (**F**). In the *Da* mutant, the left and right myotome fail to come in contact at the top of the neural tube (**H**). (**G, I**) Dorsal view of whole-mount Phalloidin (magenta) immunostaining labeling the myotome of Wt (**G**) and *Da* (**I**) embryos. Epaxial myotome is outlined, and asterisks label melanophores. The contour of the myotomes was drawn based on the Z-stack images of the dorsal myotomes to avoid ambiguity caused by melanophores. Anterior to the left. (**J**) Schematic representation of measurements to analyze the distance between the left and the right dorsal tip of the myotome (yellow) and the cross-sectional area of the dorsal myotome. For each measurement, three consecutive optical cross sections of the 10th somite were analyzed and averaged. (**K**) Distance between the left and right tip of

*Figure 1 continued on next page*

*Figure 1 continued*

the dorsal myotome 5.5 dpf – 9 dpf (n = 6 and 5 for Wt and *Da* embryos, respectively at 5.5 dpf (stage 35) (p = 0.097); n = 5 and 5 at 6 dpf (stage 36) (p = 0.075); n = 8 and 8 at 7 dpf (stage 37) (p = 0.0047); n = 5 and 6 at 8 dpf (stage 38) (p = 0.019); n = 7 and 6 at 9 dpf (stage 39) (p = 0.034); median, first and third quartiles are shown). (**L**) Cross-sectional area of the dorsal somites at 22 ss (2.25 dpf, stage 26; n = 10 somites of 5 Wt embryos, n = 12 somites of 6 *Da* embryos) (p = 0.038), 7 dpf (n = 10 somites of 5 Wt embryos, n = 10 somites of 5 *Da* embryos) (p = 0.044) and 9 dpf (n = 8 somites of 4 Wt embryos, n = 6 somites of 3 *Da* embryos) (p = 0.0019). Median, first and third quartiles are shown. HM, horizontal myoseptum; NT, neural tube. Scale bar = 50 μm. ** p < 0.01, * p < 0.05, ns, not significant.

The online version of this article includes the following figure supplement(s) for figure 1:

**Figure supplement 1.** The ventralized epaxial myotome of the *Da* mutant fails to extend sufficiently to cover the neural tube at the end of embryonic development.

## The dorsal myotome of the *Da* mutant fails to cover the neural tube

In wild-type (Wt) medaka, the dorsal ends of the myotomes first came in contact at 7 days post fertilization (dpf, stage 37) and formed the tight, thick myotome layer covering the neural tube at the end of embryonic development (9 dpf, stage 39) (*Figure 1F–G*, *Figure 1—figure supplement 1A-F*). In the ventralized *Da* mutant, however, the dorsal ends of the myotomes did not extend sufficiently and failed to cover the neural tube at the end of embryonic development (*Figure 1H–I*, *Figure 1—figure supplement 1G-L*). The ends of the ventralized dorsal myotome in the mutant displayed a round shape (not pointed as found in Wt) which resembled the morphology of the ventral myotome.

We wondered if there are other morphological differences between Wt dorsal myotome and the ventralized dorsal myotome of the *Da* mutant. Indeed, the cross-sectional area in the *Da* mutant was significantly larger compared to Wt (*Figure 1L*). Possible explanations for a larger cross-sectional area in the *Da* mutant could be a larger myofiber diameter or a higher number of myofibers, which make up the myotome. We measured the diameter of dorsal myotome muscle fibers in Wt and *Da* mutant embryos but could not observe a difference, suggesting that dorsal myotome of the *Da* mutant has a higher number of myotomal cells (*Figure 1—figure supplement 1M*).

## Proliferative activity of the dorsal DM cells is enhanced in the *Da* mutant

In fish, as in other vertebrates, the DM gives rise to muscle precursor cells, which ultimately differentiate into myofibers. In medaka the DM is a one cell-thick, Pax3/7-positive cell layer encompassing the myotome (*Figure 2A–B"*; *Hollway et al., 2007*; *Abe et al., 2019*). A high proliferative activity of the dorsal DM could explain a larger cross-sectional area of the dorsal myotome of the *Da* mutant. To test this, we performed immunohistochemistry against the mitotic marker phospho-histone H3 (pH3) and the DM marker Pax3/7 on Wt and *Da* embryos (*Figure 2C–E*). In both Wt and *Da* embryos, pH3-positive cells were randomly distributed in the dorsal DM without obvious bias (*Figure 2C–D*). At the 12-somite stage (12 ss, 1.7 dpf stage 23), when *zic1* expression in the somites becomes restricted to the dorsal region (*Kawanishi et al., 2013*), the number of pH3-positive DM cells per dorsal somite was not significantly different between Wt and *Da*. Remarkably, from 16 ss (1.8 dpf, stage 24) onwards, the number of pH3-positive cells became reduced in the Wt, whereas in *Da*, no such reduction was observed (*Figure 2E*). At 35 ss (3.4 dpf, stage 30), pH3-positive cells increased both in the Wt and *Da*, but the mutant DM cells were more proliferative (*Figure 2E*). Immunohistochemistry against another proliferation marker PCNA confirmed these findings (*Figure 2—figure supplement 1A-C*). The number of pH3-positive cells in the ventral DM was not significantly different in Wt and *Da* embryos at 22 ss (2.75 dpf, stage 26) (*Figure 2—figure supplement 2A*). These results suggest that *zic1* reduces proliferative activity of the DM, which becomes evident following the confinement of its expression to the dorsal somite region.

## Wt dorsal DM cells form numerous large, motile protrusions at the onset of dorsal somite extension

The epaxial myotome, on either side of the neural tube, extends dorsally to cover the neural tube by the end of embryonic development. In the *Da* mutant, the myotome is unable to cover the neural tube despite increased dorsal myotome growth at the hatching stage (*Figures 1L and 2*). This suggest that additionally to physical extension an active process might support the dorsal movement of somites.

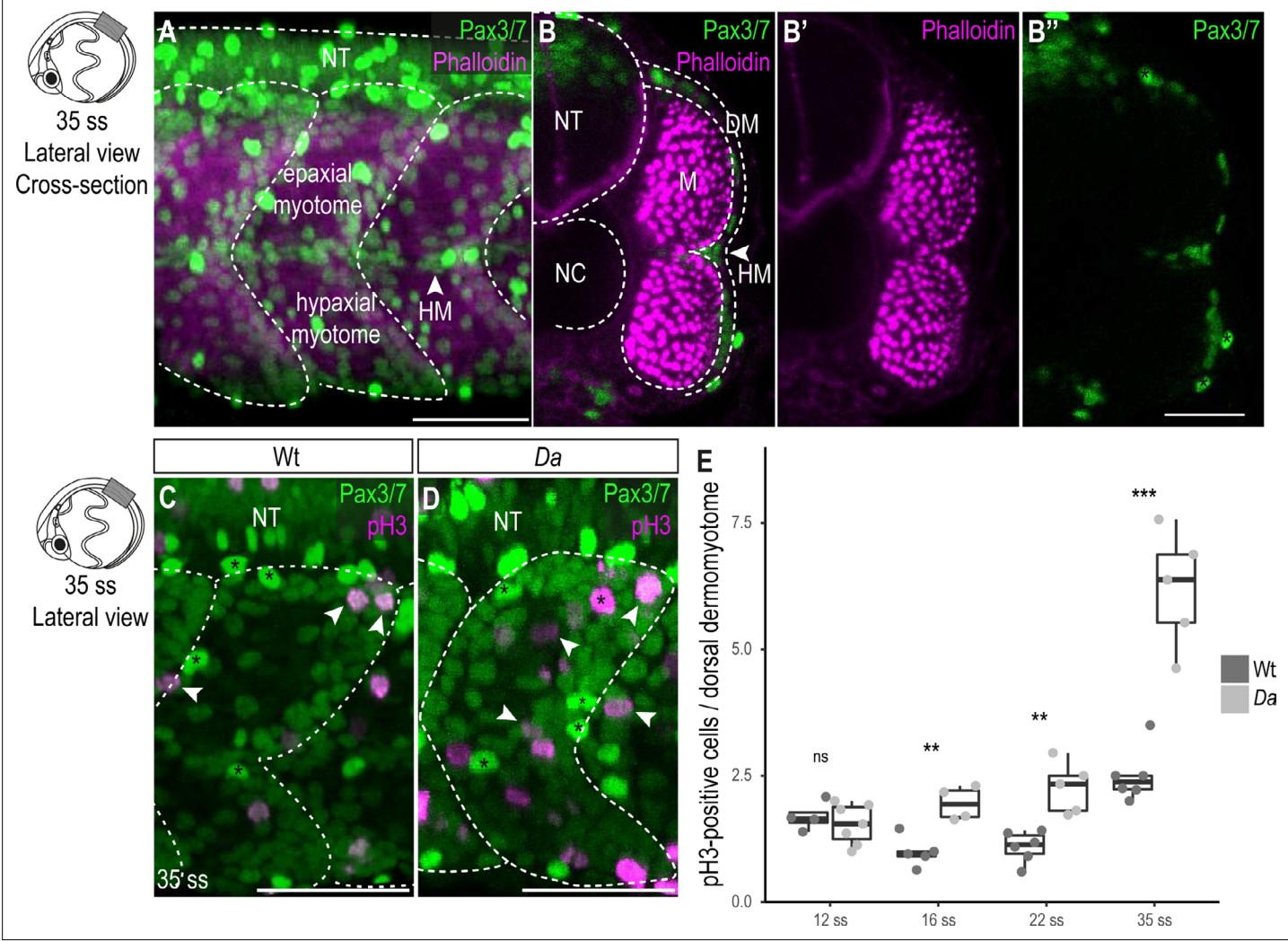

**Figure 2.** Wt dorsal DM cells show lower proliferative activity after the confinement of *zic1* expression to the dorsal somite. (**A**) Lateral view of 35 ss (3.4 dpf, stage 30) embryo, 10th somite is positioned in the center. Pax3/7 (green) labels DM cells and Phalloidin (magenta) labels myotome. The horizontal myoseptum (HM) separates the myotome into epaxial myotome (dorsal) and hypaxial myotome (ventral). (**B–B''**) Optical cross sections, myotome is labeled by Phalloidin (magenta) and encompassed by a one-cell thick layer of DM labeled with Pax3/7 (green). Asterisks mark neural crest cells which are highly Pax3/7-positive (**B''**). (**C, D**) Lateral view of Wt (**C**) and *Da* (**D**) 35 ss embryos labeled with Pax3/7 (green) and pH3 (magenta; representatives are indicated by arrowheads). Asterisks mark neural crest cells. (**E**) Quantification of pH3-positive cells in Wt and *Da* at 12 ss (n = 46.5 somites from 4 Wt embryos, n = 95 somites from 7 *Da* embryos; p = 0.48), 16 ss (n = 54.5 somites from 5 Wt embryos, n = 42.5 somites from 4 *Da* embryos; p = 0.0038), 22 ss (n = 66 somites from 6 Wt embryos, n = 40.5 somites from 5 *Da* embryos; p = 0035) and 35 ss (n = 49 somites from 6 Wt embryos, n = 47.5 somites from 5 *Da* embryos; p = 0.0008). Median, first and third quartiles are shown. ns, not significant, **p < 0.01, ***p < 0.001. Anterior to the left. Dorsal to the top. DM, dermomyotome; HM, horizontal myoseptum; M, myotome; NT, neural tube; NC, notochord. Scale bar = 50 μm.

The online version of this article includes the following figure supplement(s) for figure 2:

**Figure supplement 1.** Analysis of proliferative activity of dorsal DM cells using anti-PCNA immunohistochemistry.

**Figure supplement 2.** Difference in proliferative activity is not observed between Wt and *Da* ventral DM.

To examine the behavior of *zic1*-positive cells underlying this dorsal somite extension, we performed in vivo time-lapse imaging of dorsal somites using the transgenic line Tg(*zic1:GFP,zic4:DsRed*), which expresses GFP under the control of the *zic1* promoter and enhancers to visualize the dorsal somitic cells (*Kawanishi et al., 2013*) (hereafter called Tg(*zic1:GFP*) since the DsRed fluorescence was negligible in the following analyses). Intriguingly, around 22 ss and onwards, cells at the tip of the dorsal somites started to form numerous large protrusions extending dorsally towards the top of the neural tube (*Figure 3A, Video 1*). We defined the beginning of protrusion formation as the onset of dorsal somite extension. Close-up views of the time-lapse images (*Figure 3A, Video 1*) showed that

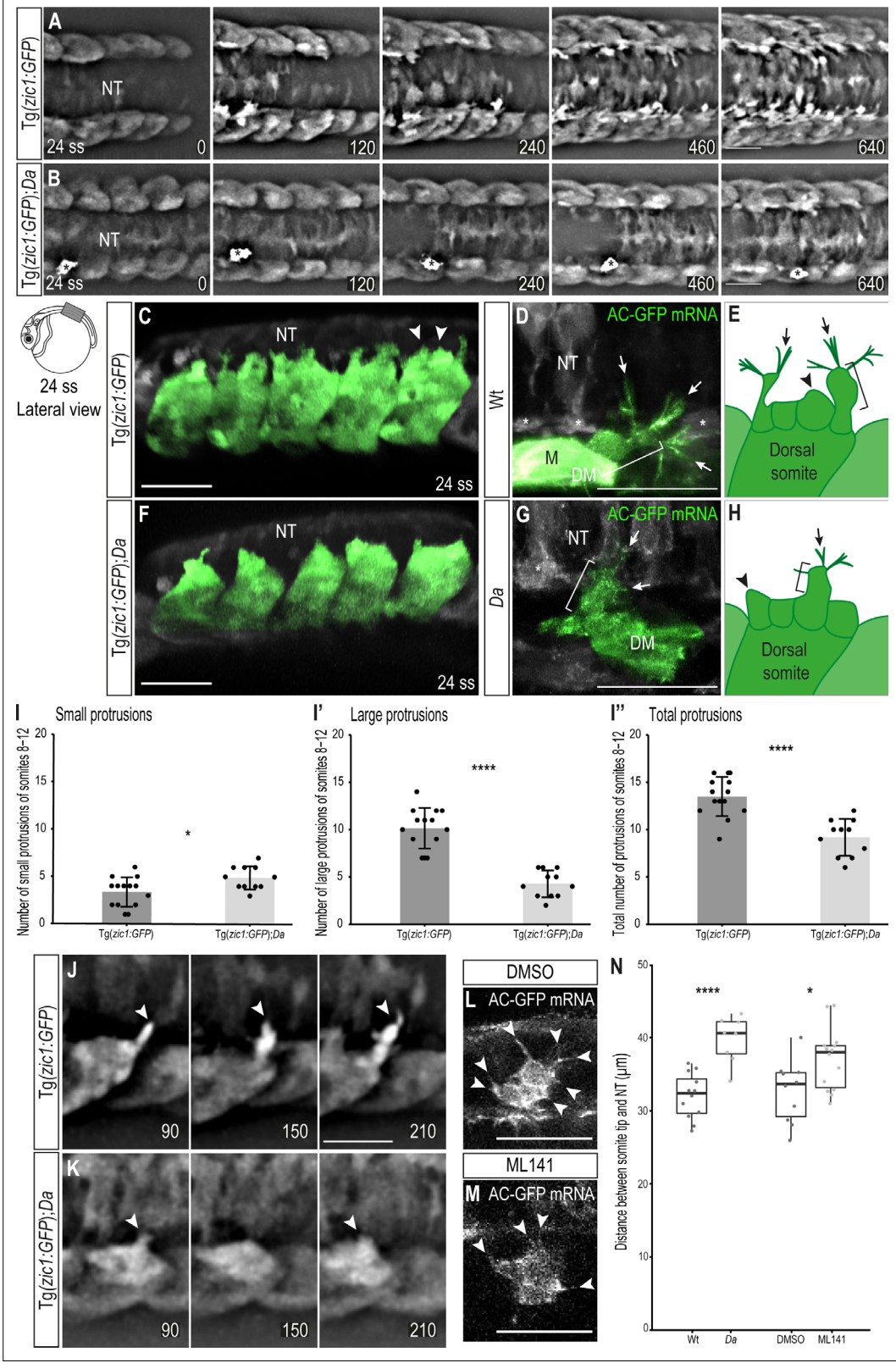

**Figure 3.** Wt dorsal DM cells form numerous large, motile protrusions at the onset of somite extension. (**A, B**) Dorsal view of time-lapse in vivo imaging of onset of dorsal somite extension at 24 ss (2.4 dpf, stage 27) of Tg(*zic1:GFP*) (**A**) and Tg(*zic1:GFP);Da* (**B**) embryos. 15th somite is positioned in the center, z-stacks were imaged every 10 min, time is displayed in min. Asterisks indicate migrating melanophore. Scale bar = 50 µm. (**C, F**) Lateral

*Figure 3 continued on next page*

*Figure 3 continued*

view of in vivo imaging of Tg(*zic1:GFP*) (**C**) and Tg(*zic1:GFP*);*Da* (**F**). Signals in dorsal somites are highlighted in green. Arrowheads indicate small protrusions, arrow indicates large protrusion. Scale bar = 50 μm. (**D, G**) Dorsal view of in vivo imaging of large protrusion in Wt (**D**) and *Da* (**G**). Embryos were injected with Actin-Chromobody GFP (AC-GFP) mRNA. Signals in somitic cells are highlighted in green. Brackets indicate a lamellipodia-like structure. Arrows indicate filopodia branching out from dorsal tips of lamellipodia-like core. Asterisks indicate sclerotome cells. Scale bar = 25 μm. (**E, H**) Summary of dorsal DM cell protrusions in Wt (**E**) and *Da* (**H**) embryos. Arrowheads indicate small protrusions, brackets indicate lamellipodia-like core structure of large protrusions, and arrows indicate filopodia bundles branching off from dorsal tips of large protrusions. (**I-I″**) Quantification of protrusions from Tg(*zic1:GFP*) (n = 14 embryos) and Tg(*zic1:GFP*);*Da* (n = 11 embryos). Protrusions of the 8th-12th somite were counted (mean ± SD, p = 0.01 for small protrusions, p = 3.3e-08 for large protrusions, p = 2.1e-05 for total protrusions). (**J–K**) Lateral view of protrusions extracted from time-lapse imaging of (**A–B**). Arrowheads indicate tip of protrusions, time is displayed in min. (**J**) Protrusion observed in Tg(*zic1:GFP*). (**K**) Protrusion observed in Tg(*zic1:GFP*);*Da*. Scale bar = 25 μm. (**L, M**) Z-planes of large protrusions of 24 ss embryos treated with DMSO (**L**) or ML141 (**M**). Arrowheads indicate filopodia. Scale bar = 25 μm. (**N**) Distance between dorsal somite tip and top of neural tube in 24 ss embryos (n = 12 somites from 6 Wt embryos, n = 8 somites from 4 *Da* embryos, p = 6.8e-05; n = 10 somites from 5 embryos treated with DMSO, n = 14 somites from 7 embryos treated with ML141, p = 0.021). Median, first and third quartiles are shown. ****p < 0.0001, *p < 0.5. Anterior to the left. DM, dermomyotome; M, myotome; NT, neural tube.

The online version of this article includes the following figure supplement(s) for figure 3:

**Figure supplement 1.** Protrusion forming cells at the tips of the dorsal somites are DM cells.

**Figure supplement 2.** Dorsal DM cells form protrusions throughout dorsal somite extension.

these protrusions were motile and dynamically formed new branches at their dorsal tips (*Figure 3J*, *Figure 3—figure supplement 1D*). Immunohistochemistry revealed that the protrusion-forming cells belong to the DM (*Figure 3—figure supplement 1A-A″*).

To characterize the protrusions, we classified them according to their length into small ( < 8 μm, *Figure 3C* arrowheads) and large ( ≥ 8 μm, *Figure 3C* arrow) protrusions (*Figure 3—figure supplement 1B-C*). Based on their shape, we reasoned that the small protrusions correspond to lamellipodia (*Figure 3C*, arrowheads), while the large protrusions appeared more complex. To investigate the nature of large protrusions, we injected Actin-Chromobody GFP mRNA to visualize the actin skeleton. The large protrusions were found to exhibit a complex architecture consisting of a lamellipodia-like core structure (*Figure 3D*, bracket) with additional multiple bundles of filopodia (protrusions with

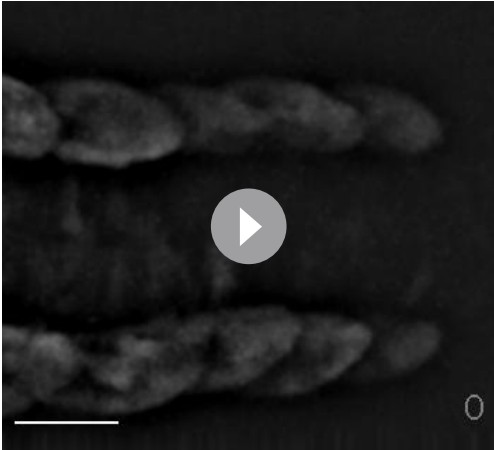

**Video 1.** Onset of dorsal somite extension in Tg(*zic1:GFP*). Dorsal view of time-lapse in vivo imaging of 24 ss Tg(*zic1:GFP*) embryo. 15th somite is positioned in the center, z-stacks were imaged every 10 min, time is displayed in min. Anterior to the left. Scale bar = 50 μm.

https://elifesciences.org/articles/71845/figures#video1

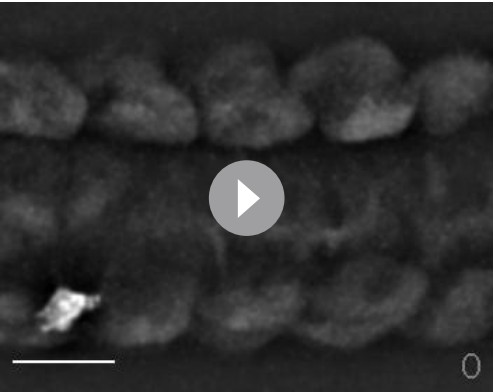

**Video 2.** Onset of dorsal somite extension in Tg(*zic1:GFP*);*Da*. Dorsal view of time-lapse in vivo imaging of 24 ss Tg(*zic1:GFP*);*Da* embryo. 15th somite is positioned in center, z-stacks were imaged every 10 min, time is displayed in min. Bright cell at the bottom migrating to the right is a melanophore. Anterior to the left. Scale bar = 50 μm.

https://elifesciences.org/articles/71845/figures#video2

linear arranged actin filaments) branching out from their dorsal tips (*Figure 3D*, arrows, summarized in *Figure 3E*).

Interestingly, time-lapse in vivo imaging of Tg(*zic1:GFP*);*Da* showed that in the *Da* background, protrusions started to form later (*Figure 3B*, *Video 2*) and the number of large protrusions and protrusions in total was significantly lower than in Wt (*Figure 3I–I"*). In addition, protrusions were transient and mostly failed to form new branches at their dorsal tips (*Figure 3K*, *Figure 3—figure supplement 1E*). While no difference in the actin skeleton of small protrusions could be observed, filament bundles branching out from large protrusions of *Da* DM cells contained fewer and shorter filopodia compared to Wt (*Figure 3G*, arrowheads, summarized in *Figure 3H*). These results indicate that the protrusive activity, especially the ability to form large protrusions, is significantly reduced in the *Da* mutant. The large protrusions of the dorsal DM cells continuously appeared at later stages of dorsal somite extension, too (*Figure 3—figure supplement 2*).

To investigate the role of large protrusions during the onset of dorsal somite extension, we inhibited filopodia formation using ML141 (*Figure 3L and M*). ML141 specifically inhibits Cdc42/Rac1 GTPases, which are critical for filopodia formation (*Hong et al., 2013*; *Fantin, 2015*). Intriguingly, dorsal somites of embryos treated with ML141 extended significantly less dorsally compared to control embryos treated with DMSO (*Figure 3N*). Taken together, our data suggest that the unique large protrusions of the dorsal DM are involved in guiding the epaxial myotome dorsally and *zic1* might promote this function.

## DM cells delaminate and accumulate between opposing somites during the late phase of dorsal somite extension

We continued to trace the behavior of the DM tip cells until the somites reach the top of the neural tube. Indeed, in vivo imaging of Tg(*zic1:GFP*) revealed that dorsal DM cells continued to form protrusions, and additionally, some of them delaminated to become Zic1-positive mesenchymal cells accumulating in the space between the dorsal ends of the left and the right somites (*Figure 4A*, star-shaped cells, arrowheads) from 4.5 dpf (stage 33) onwards. This is consistent with previous observations of strongly *zic1* expressing mesenchymal cells in Wt at late embryonic stages (*Ohtsuka et al., 2004*). As dorsal somite extension proceeded, the number of these mesenchymal cells increased, filling the space between the two myotomes (*Figure 4A–C*, arrowheads, *Videos 3–4*, arrowheads indicate representative mesenchymal cells). These mesenchymal cells formed protrusions towards neighboring mesenchymal cells and DM cells at the tip of somites, creating a dense cellular network between the dorsal ends of the somites. Mosaic cell-labelling demonstrated that the mesenchymal cells originated from the DM (*Figure 4—figure supplement 1A-F*, arrowheads). Interestingly, while mesenchymal cells dynamically formed protrusions, they showed no extensive migratory behavior and were rather stationary (*Videos 3–4*, *Figure 4—figure supplement 2*). This could suggest that these protrusions fulfill a non-migratory function. When the opposing somites came in contact with each other at 8 dpf (stage 38, 1 day before hatching), the mesenchymal cells tended to attach to the nearest DM cells at the tip of the somite, bridging the gap between the left and right DM cells (*Figure 4I*, *Figure 4—figure supplement 1M-M''''*).

In *Da* mutants, mesenchymal DM cells were also detected in the space between the two myotomes, but the timing of their appearance was delayed, that is between 5 dpf (stage 34) and 5.5 dpf (stage 35) (*Figure 4D–F*) (4.5 dpf in Wt). Additionally, *Da* mesenchymal cells exhibited a rounder morphology and formed significantly fewer protrusions compared to Wt while the number of mesenchymal cells was not largely affected (*Figure 4G and H*, *Figure 4—figure supplement 1G-L*).

To examine the function of mesenchymal DM cells, we ablated these cells between the 10th somites of 5.5 dpf Tg(*zic1:GFP*) embryos using an UV laser (*Figure 4J–K'*, *Figure 4—figure supplement 3A-B'*). Intriguingly, the distance between the left and the right tips of the 10th somites increased after ablating the mesenchymal DM cells while the neighboring 9th somites continued to shorten the gap (*Figure 4L*). This suggests that mesenchymal DM cells hold the left and the right somite together to promote the dorsal somite extension at this late phase of myotome development. After 16 hr post ablation, dorsal extension of the 10th somites eventually resumed as the mesenchymal DM cells regenerated at the ablation site (*Figure 4L*).

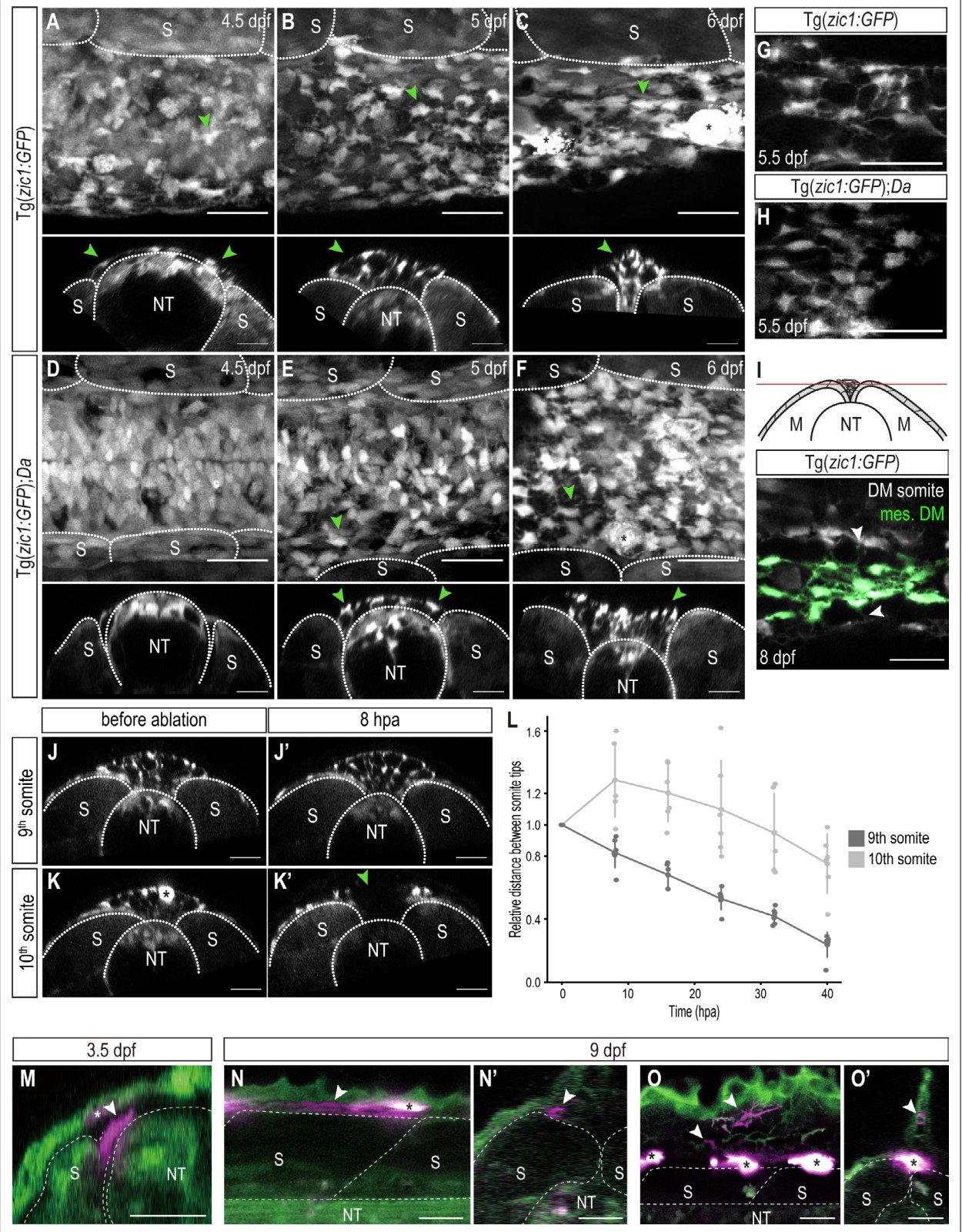

**Figure 4.** DM cells delaminate from the dorsal somite at the end of dorsal somite extension and accumulate between somites. (**A–F**) Dorsal (top) and cross-sectional (bottom) views of Tg(*zic1:GFP*) at 4.5 dpf (**A**), 5 dpf (**B**), and 6 dpf (**C**) and Tg(*zic1:GFP*);*Da* at 4.5 dpf (**D**), 5 dpf (**E**) and 6 dpf (**F**). 10th somite is positioned in center. Arrowheads point to exemplary mesenchymal DM cells, and asterisks mark melanophores. (**G, H**) Dorsal view of mesenchymal DM cells of Tg(*zic1:GFP*) (**G**) and Tg(*zic1:GFP*);*Da* (**H**). (**I**) Z-plane of dorsal view of 10th somite of Tg(*zic1:GFP*) embryo. Mesenchymal DM (mes. DM)

*Figure 4 continued on next page*

*Figure 4 continued*

cells are colored green. Arrowheads indicate exemplary protrusions formed between somitic DM cell and mesenchymal DM cell. (**J-K′**) Cross-sections of 9th somite before ablation (**J**), 8 hr post ablation (hpa) (**J′**), and 10th somite before ablation (**K**) and 8 hpa (**K′**) of Tg(*zic1:GFP*) 5.5 dpf embryo. Green arrowhead indicates the ablation site, and asterisk indicates a pigment cell. (**L**) Quantification of relative distance between the left and the right tips of the 9th and 10th somites after laser ablation (n = 6 Tg(*zic1:GFP*) embryos). (**M-O′**) Fate mapping, using KikGR-mediated photoconversion, of dorsal DM cells with protrusions. (**M**) A single dorsal DM cell at the dorsal tip of a somite (arrowhead) was labeled at 3.5 dpf. Cross-section of the trunk is shown. Asterisk indicates an ectopically labeled epidermis cell. (**N-O′**) At hatching stage (9 dpf), descendants of the labeled DM cell contribute to blood vessels (arrowheads in N, **N′**) and fin mesenchyme in the dorsal fin fold (arrowheads in O, **O′**). (**N**) and (**O**), lateral views; (**N′**) and (**O′**), cross sections. Asterisks indicate autofluorescent pigment cells. Anterior to the left. M, myotome; NT, neural tube; S, somite. Scale bar = 25 μm.

The online version of this article includes the following figure supplement(s) for figure 4:

**Figure supplement 1.** Mesenchymal DM cells originate from dorsal somites.

**Figure supplement 2.** Mesenchymal DM cells do not migrate significantly.

**Figure supplement 3.** Function and fate of mesenchymal DM cells.

Collectively, dorsal DM cells and the mesenchymal cells derived from them seem to actively participate in the entire process of dorsal somite extension, from its onset to neural tube coverage at the end.

Finally, we examined the fate of the dorsal DM cells that derive the mesenchymal cells. We employed a photoconversion technique mediated by a photoconvertible protein KikGR to specifically label a single dorsal DM cell exhibiting protrusions at the tip of a 10th and 20th somites during dorsal somite extension (*Figure 4M*, arrowhead) and tracked them until the hatching stage. We found that the labeled dorsal DM cells eventually differentiated into cells surrounding blood vessels (likely mural cells, n = 8/9 and 4/8 for 10th and 20th somites, respectively; *Figure 4N and N′*, arrowheads) and mesenchymal cells in the dorsal fin fold (n = 4/8 for 20th somites; *Figure 4O and O′*, arrowheads); however, we did not observe any labeled axial muscles (n = 0/9 and 0/8 for 10th and 20th somites, respectively). This suggests that the dorsal DM cells at the tip of somites, guiding myotome extension, do not become myotomal cells themselves at least until the end of embryonic development (hatching stage, 9 dpf).

## Zic1 regulates the expression of dorsal-specific genes during somite differentiation

We then addressed the molecular machinery controlling the dorsal somite extension investigated above. Since somite extension is impaired in the *zic1*-enhancer mutant *Da* (*Moriyama et al., 2012*; *Kawanishi et al., 2013*), we reasoned that downstream genes of Zic1 are regulators of this process.

First, we identified genes which are specifically expressed in the dorsal somites. For this, we collected somites of the transgenic line Tg(*zic1:GFP*) by removing other tissues after pancreatin treatment, FACS sorted them into the dorsal (GFP+)

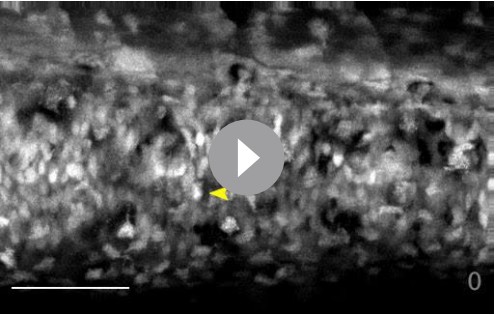

**Video 3.** Mesenchymal DM cells during dorsal somite extension at 4.5 dpf. Dorsal view of time-lapse in vivo imaging of 4.5 dpf Tg(*zic1:GFP*) embryo. 10th somite positioned in center, z-stacks were imaged every 10 min, time is displayed in min. Arrowhead indicates representative mesenchymal DM cell. Anterior to the left. Scale bar = 50 μm.

https://elifesciences.org/articles/71845/figures#video3

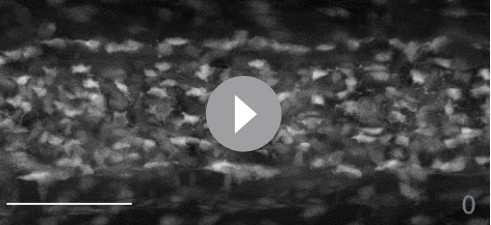

**Video 4.** Mesenchymal DM cells during dorsal somite extension at 5.5 dpf. Dorsal view of time-lapse in vivo imaging of 5.5 dpf Tg(*zic1:GFP*) embryo. 10th somite is positioned in center, z-stacks were imaged every 10 min, time is displayed in min. Arrowhead indicates representative mesenchymal DM cell. Anterior to the left. Scale bar = 50 μm.

https://elifesciences.org/articles/71845/figures#video4

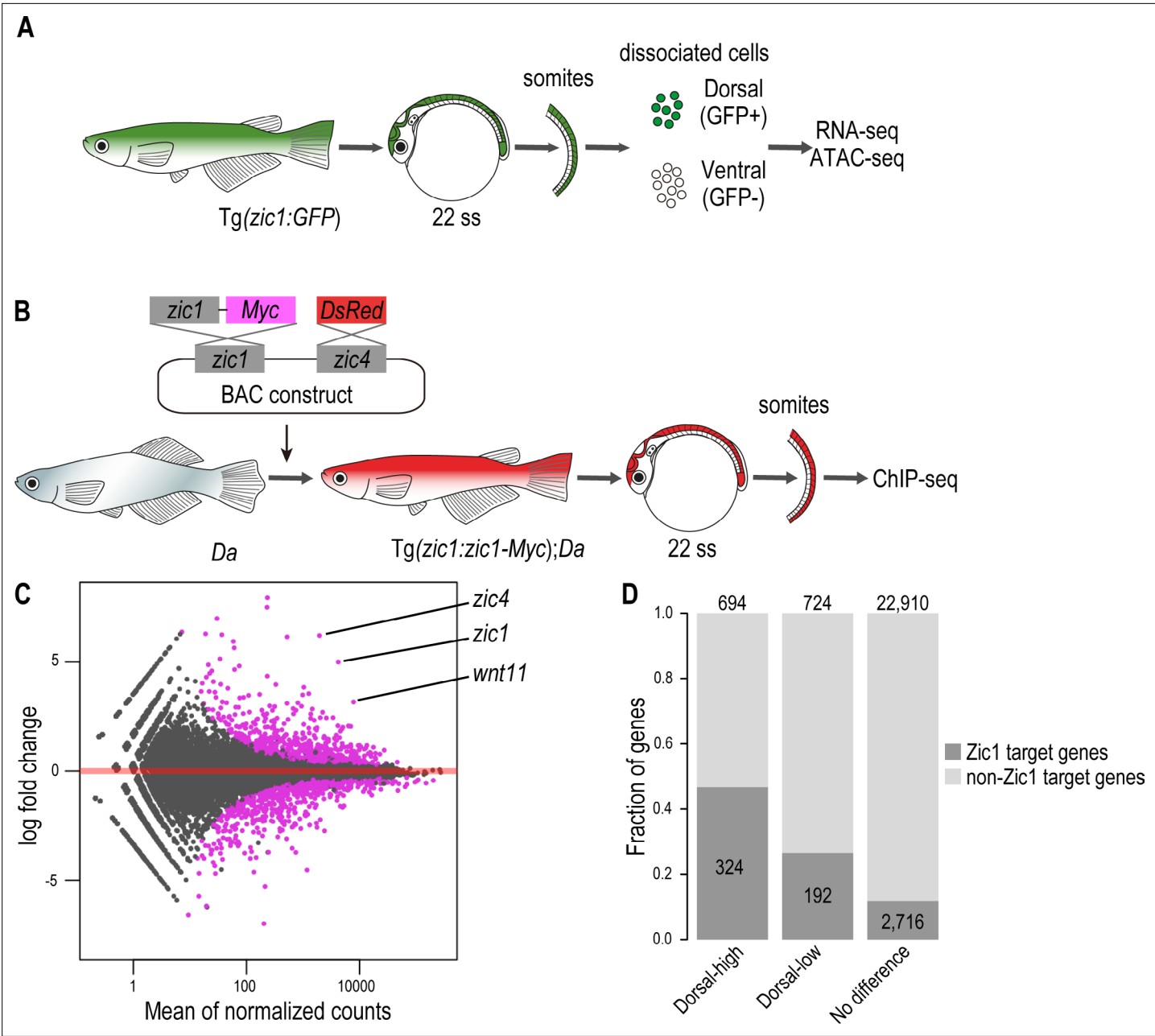

**Figure 5.** Zic1 regulates dorsal-specific expression of genes in the somites. (**A**) Schematic representation of preparation of dorsal and ventral somite cells for RNA-seq and ATAC-seq. (**B**) Schematic representation of generating the transgenic line Tg(*zic1:zic1-Myc*);*Da*. Somites expressing *zic1-Myc* are subjected to ChIPmentation against Myc. (**C**) Analysis of RNA-seq revealed that 694 genes are expressed specifically in the dorsal somites (among these genes are *zic1*, *zic4* and *wnt11*) and 724 genes are expressed specifically in ventral somites. Magenta indicates differentially expressed genes (adjusted p-value < 0.01). (**D**) The ChIPmentation against Zic1 revealed that 324 of dorsal-high genes, 192 dorsal-low genes and 2716 non differential expressed genes in the somites are potential Zic1 target genes.

The online version of this article includes the following figure supplement(s) for figure 5:

**Figure supplement 1.** In Tg(*zic1:zic1-Myc*);*Da* fish, the ventralized trunk phenotype is rescued.

**Figure supplement 2.** Common ZIC family DNA motifs are enriched in Zic1 peaks.

and ventral (GFP-) somitic cells including the DM cells, and performed RNA-seq and ATAC-seq on both cell populations (*Figure 5A*, *Figure 5—figure supplement 1A*). The RNA-seq identified 1,418 differentially expressed genes. Among them 694 genes showed higher expression in the dorsal somites (termed hereafter dorsal-high genes), and 724 genes showed higher expression in the ventral

somites (termed hereafter dorsal-low genes) (*Figure 5C*). We confirmed that *zic1* and *zic4* were found among the dorsal-high genes (*Figure 5C*, *Supplementary file 1*).

Next, we identified potential direct Zic1-target genes by investigating Zic1 binding sites. Since there were no suitable antibodies available to perform ChIP-seq against medaka Zic1, we created a transgenic line expressing a Myc tagged *zic1* in the *Da* background under the control of *zic1* promoter and enhancers (Tg(*zic1:zic1-Myc,zic4:DsRed*);*Da*, called *Tg(zic1:zic1-Myc);Da* hereafter). This transgene (*zic1:zic1-Myc*) efficiently rescued the ventralized phenotype (dorsal and anal fin shape, pigmentation pattern, and body shape) of the *Da* mutant (*Figure 5—figure supplement 1B-C*), verifying the full functionality of the tagged Zic1 protein. Somites of the transgenic line *Tg(zic1:zic1-Myc);Da* were dissected and subjected to ChIP-seq using antibodies against Myc (*Figure 5B*). Since somites contain a low number of cells, we applied ChIPmentation (*Schmidl et al., 2015*) to identify genome-wide Zic1 binding sites. From two biological replicates, 5247 reliable ChIP peaks were identified, and we confirmed that the enriched DNA motif among these peaks was showing high similarity with previously identified binding motifs of ZIC family proteins (*Figure 5—figure supplement 2A-B*). Then, we associated each Zic1 peak to the nearest gene within 50 kb and identified 3232 genes as Zic1 target genes. By comparing the ChIPmentation with RNA-seq data, we found that Zic1 target genes are overrepresented in dorsal-high and dorsal-low genes. While 47% of the dorsal-high genes and 27% of the dorsal-low genes were found to be Zic1 target genes, only 12% of genes which showed no differential expression in dorsal or ventral somites were potential Zic1 downstream target genes (*Figure 5D*, *Supplementary file 1*). This suggests that Zic1 can function as transcriptional activator and repressor of versatile genes, but the former role seems to be dominant.

## *wnt11* is a direct downstream target of Zic1 and down-regulated in the dorsal somites of the *Da* mutant

To identify potential regulators of dorsal somite extension, we further investigated the differentially expressed Zic1 target genes. Gene Ontology (GO) analysis indicated that both dorsal-high and dorsal-low gene groups, regardless of whether they are Zic1 targets or non-Zic1 targets, were significantly enriched in development related GO terms (*Figure 6A*, *Figure 6—figure supplement 1A*, *Supplementary files 2 and 3*). This indicates that Zic1 regulates a number of developmental genes both directly and indirectly. These results are consistent with the fact that Zic1 regulates various dorsal-specific morphologies of somite-derivatives (*Kawanishi et al., 2013*).

In the dorsal-high Zic1 targets, GO terms related to cell migration showed higher enrichment (e.g. 'chemotaxis' (p = 5.62E-10), 'locomotion' (6.65E-15), 'ameboidal-type cell migration' (p = 9.32E-10)) than dorsal-low genes or non-Zic1 target genes (*Supplementary file 3*). Interestingly, Wnt signaling pathway genes (e.g.: *axin2, wnt11, sp5, lrp5, fzd10, prickle1a*) and semaphorin-plexin signaling pathway genes (e.g.: *sema3a, sema3c, sema3g, plxna1, plxna2, plxnb2, plxnb3, nrp2*) were included in these gene groups. Indeed, the GO terms 'Wnt signaling pathway' and 'Semaphorin-plexin signaling pathway' themselves were also significantly enriched (p = 4.45E-7 and 1.94E-8, respectively) in dorsal-high Zic1 target genes (*Supplementary file 2*). This suggests that Zic1 directly regulates Wnt pathway, semaphorin and plexin genes, possibly to regulate cell movement in the dorsal somites. We also noticed that the terms 'extracellular matrix organization' (p = 7.74E-7; e.g.: *adamts20, fbln1*), 'cell communication' (p = 1.95E-5; e.g.: *efna5, epha3*) are enriched, suggesting that these genes also affect dorsal somite cell behavior.

Wnt signaling pathway components also ranked high among the dorsal-high Zic1 target genes by pathway enrichment analysis (*Figure 6B*, *Figure 6—figure supplement 1B*). Among them, we focused on *wnt11* for further analyses due to the following reasons: First, *wnt11* was one of the most differentially expressed genes in dorsal somites (*Figure 5C*, *Supplementary file 1*). Second, previous studies implicated Wnt11 in protrusion formation and cell migration (*Ulrich et al., 2003*; *De Calisto et al., 2005*; *Garriock and Krieg, 2007*; *Matthews et al., 2008*). This is particularly interesting since DM cells also exhibit protrusions and migration activity during dorsal somite extension, which are defective in *Da* mutants.

At the *wnt11* locus, peaks of the Zic1-ChIP overlapped with intergenic open chromatin regions downstream of *wnt11*. These sites were more accessible in dorsal somites than in ventral somites, suggesting that Zic1 regulates *wnt11* directly (*Figure 6C*). Additionally, in situ hybridizations against *wnt11* performed on Wt and *Da* embryos indicated that *wnt11* expression is significantly reduced at

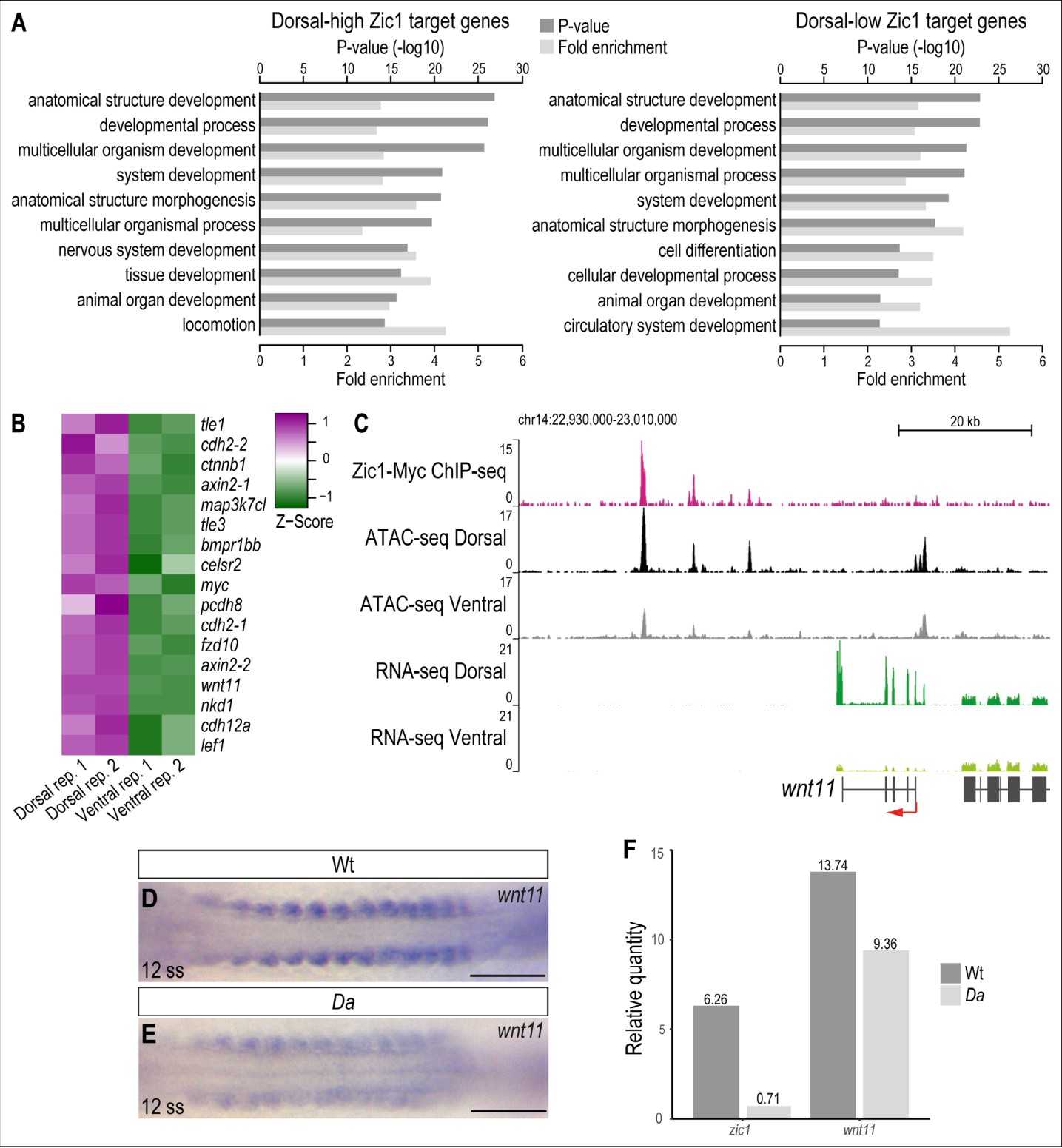

**Figure 6.** *wnt11* is a direct downstream target of Zic1 and down-regulated in the dorsal somites of the *Da* mutant. (**A**) GO analysis of dorsal-high (left) and dorsal-low (right) Zic1 target genes. (**B**) Pathway enrichment analysis indicated that genes associated with Wnt signaling pathway are specifically enriched in dorsal-high Zic1 target genes. (**C**) Analysis of the *wnt11* locus. Peaks form the ChIPmentation against Zic1 (magenta track) overlap with open chromatin regions in the dorsal somites (black track), while this genomic region is less open in ventral somites (grey track). RNA-seq data revealed that *wnt11* is highly expressed in dorsal somites (dark green track) and minimally in ventral somites (light green track). (**D, E**) Dorsal view of tails of whole-mount in situ hybridization against *wnt11* performed in Wt (**D**) and *Da* (**E**) 12 ss embryos. The expression of *wnt11* is reduced in the *Da* mutant. (**F**) RT-

*Figure 6 continued on next page*

*Figure 6 continued*

PCR performed on pooled tails of Wt and *Da* 12 ss embryos indicates that *zic1* expression is reduced by 8.8-fold and *wnt11* expression by 1.4-fold in the *Da* mutant. Anterior to the left. Scale bar = 100 μm.

The online version of this article includes the following figure supplement(s) for figure 6:

**Figure supplement 1.** Wnt signaling pathway is enriched in dorsal-high Zic1 target genes.

the dorsal tip of *Da* dorsal somites including the DM (*Figure 6—figure supplement 1C-D*). Already when the *zic1* expression becomes restricted in the dorsal somites (12 ss) and with proceeding development when *zic1* expression gets further restricted to the most dorsal part of the somites and mesenchymal DM cells (*Ohtsuka et al., 2004*), the expression of *wnt11* in the dorsal somites of the *Da* mutant was reduced compared to Wt (*Figure 6D–F*, *Figure 6—figure supplement 1C-F'*).

Taken together, we identified *wnt11* as promising downstream target gene of Zic1, which could be a novel somite dorsalization factor and play a role in dorsal somite extension.

## Wnt11 regulates cell behavior of dorsal DM

Our previous RNA-seq dataset revealed that the expression of *wnt11* starts before gastrulation and increases as development proceeds (*Figure 7—figure supplement 1A*; *Nakamura et al., 2021*). Injection of single guide RNAs (sgRNAs) against *wnt11* to knock out the gene revealed that the resultant genetically mosaic F0 embryos displayed delayed epiboly movement and subsequent morphological defects, including shorter body axis and impaired trunk development (*Figure 7—figure supplement 2*), implying the difficulty of assessment of the *wnt11* function at later stages using the *wnt11* mutant. To circumvent the problem, we took two different approaches to knock-down *wnt11* during dorsal somite extension. Firstly, we used a *wnt11* anti-sense morpholino (*wnt11* MO) and determined a concentration which resulted in maximal knock-down effects during dorsal somite extension with minimal gastrulation phenotypes (see Materials and methods). Secondly, we performed temporally controlled knock-down of *wnt11* after gastrulation using photo-cleavable Photo-Morpholinos (PhotoMOs) (*Tallafuss, 2012*; *Figure 7A*). Strikingly, in *wnt11* Photo-Morphants with severe phenotypes the dorsal myotome failed to cover the neural tube (n = 7, *Figure 7B–C*), a phenotype similar to the *Da* mutant myotome. Additionally, the myofibers of the Photo-Morphants at 9 dpf were shorter and less organized. This phenotype is consistent with a previously reported role of Wnt11 during early myogenesis, where it regulates the elongation and orientation of myoblasts (*Gros et al., 2009*).

Next, we investigated the proliferative activity of dorsal DM cells in *wnt11* morphants and found that knock-down of *wnt11* (either by conventional MO or by PhotoMO) induced significantly more pH3-positive cells per somite, compared to control embryos at 22 ss (*Figure 7D*). These findings are similar to the observations previously made in the dorsal DM of *Da* mutants, and suggest that Wnt11 is negatively regulating proliferation of dorsal DM cells.

We then explored whether the onset of dorsal somite extension in *wnt11* morphants is similarly impaired as in the *Da* by time-lapse in vivo imaging. Remarkably, we observed a protrusion formation behavior of *wnt11* morphant DM cells similar to *Da* DM cells, namely delayed onset of protrusion formation and the formation of fewer, shorter protrusions (*Figure 7E*, *Video 5*). Quantification of protrusions indicated that *wnt11* morphants have significantly fewer large protrusions and protrusions in total, compared to control embryos (*Figure 7F–H"*).

Overall, knock-down of the Zic1 target gene *wnt11* recapitulated the phenotype of *Da* DM cells (*Figure 3G–G"*), suggesting the crucial role of Wnt11 in regulating protrusion formation of DM cells.

To further confirm the importance of Wnt11 during dorsal somite extension, we performed rescue experiments in *Da* embryos. At 18 ss (2.1 dpf, stage 25), we injected a mix of human recombinant Wnt11 (hrWnt11) protein (or BSA in the control group) and Dextran Rhodamine onto the top of the 10th somite of Tg(*zic1:GFP*);*Da* embryos (*Figure 8A–B"*). Strikingly, the dorsal DM of *Da* embryos injected with Wnt11 formed significantly more large protrusions and more protrusions in total compared to *Da* embryos injected with BSA only (*Figure 8C–C"*). This indicates that the ventralized *Da* DM protrusion phenotype can be partially rescued by Wnt11 protein injections and further emphasizes the importance of Wnt11 in somite dorsalization and during dorsal somite extension.

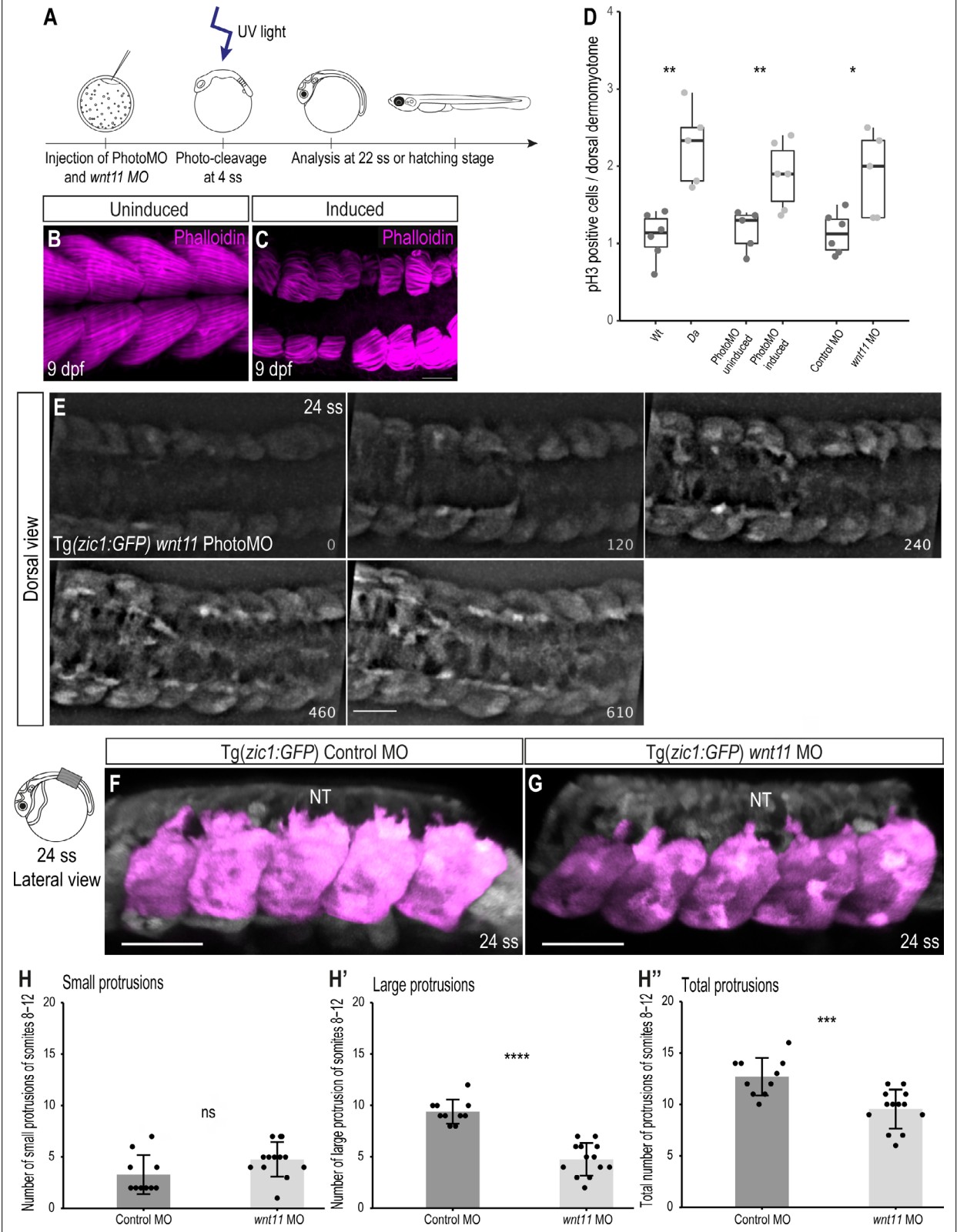

**Figure 7.** Knock-down of *wnt11* in Wt embryos recapitulates the *Da* dorsal somite phenotype. (**A**) Schematic outline of PhotoMO mediated knock-down of *wnt11*. (**B, C**) Dorsal view of maximum projection of whole-mount Phalloidin staining of 9 dpf embryos injected with PhotoMO but not photocleaved (**B**) and photo-cleaved (**C**). The epaxial myotomes of the *wnt11* morphant are affected. (**D**) Quantification of pH3-positive cells in the dorsal DM of 22 ss embryos. Embryos with reduced levels of *wnt11* in their dorsal somites (n = 40.5 somites from 5 *Da* embryos (adopted from *Figure 2E*), n = 61.5

*Figure 7 continued on next page*

*Figure 7 continued*

somites of 6 *wnt11* PhotoMO-Morphant embryos and n = 52 somites from 5 *wnt11* morphant embryos) have significantly more pH3-positive cells in their dorsal DM compared to the respective controls (n = 66 somites from 6 Wt embryos (adopted from *Figure 2E*), n = 51 somites of 5 uninduced PhotoMO embryos and n = 65 somites from 6 Control MO embryos; p = 0.0035, 0.0091, 0.021, respectively). Median, first and third quartiles are shown. (**E**) Dorsal view of 24 ss Tg(*zic1:GFP*) embryo injected with *wnt11* PhotoMO and photo-cleaved at 4 ss. Z-stacks were recorded every 10 min, time is displayed in min, and 15[th] somite is positioned in the center. (**F, G**) Lateral view of dorsal somites (colored magenta) of Tg(*zic1:GFP*) injected with Control MO (**F**) and *wnt11* MO (**G**). (**H-H″**) Quantification of protrusions formed by the 8[th]-12[th] somite of Tg(*zic1:GFP*) injected with Control MO (n = 10 embryos) or *wnt11* MO (n = 13 embryos) (mean ± SD, p = 0.069 for small protrusions, p = 7.7e-08 for large protrusions, p = 0.00064 for total protrusions). Anterior to the left. NT, neural tube. Scale bar = 50 µm. *p < 0.05, **p < 0.01, ***p < 0.001, ****p < 0.0001, ns, not significant.

The online version of this article includes the following figure supplement(s) for figure 7:

**Figure supplement 1.** *wnt11* expression starts before gastrulation and increases with proceeding development.

**Figure supplement 2.** Injection of *wnt11* sgRNAs results in delayed epiboly movement, and impaired body axis and trunk development.

## Wnt11 acts through the Wnt/Ca²⁺ signaling pathway at the onset of somite extension

Finally, we investigated through which signaling pathway the non-canonical Wnt11 acts. In *Xenopus*, Wnt11 acts through the Wnt/Ca²⁺ pathway regulating the migration of cells from the dorsal somite and the neural crest into the dorsal fin fold (*Garriock and Krieg, 2007*). To examine whether this signaling pathway also plays a role during dorsal somite extension, we inhibited the Wnt/Ca²⁺ signaling pathway using KN-93 in Tg(*zic1:GFP*) embryos from 4 ss (1.3 dpf, stage 20)–22 ss (*Figure 9A*). KN-93 specifically inhibits CaMKII, a component of the Wnt/Ca²⁺ pathway, by binding to it and preventing its phosphorylation, thus keeping it in an inactivated state (*Sumi et al., 1991*; *Tombes et al., 1995*; *Wu and Cline, 1998*; *Garriock and Krieg, 2007*; *Rothschild et al., 2013*). Embryos treated with KN-93 showed a higher number of pH3-positive dorsal DM cells per somite, compared to embryos in the control group (*Figure 9B*). Furthermore, DM cells at the tip of the dorsal somite of embryos treated with KN-93 formed significantly fewer large and fewer protrusions in total, compared to embryos in the control group (*Figure 9C–C″*), although the effect was less significant compared to the *wnt11* morphants (*Figure 7*).

From these results, we suggested that Wnt11 potentially acts through the Wnt/Ca²⁺ signaling pathway at the onset of somite extension. Wnt/Ca²⁺ signaling pathway is known to regulate actin polymerization (*Choi and Han, 2002*; *Kohn and Moon, 2005*) which could explain the dynamic protrusive activity of the DM cells (*Figure 3*), emphasizing the importance of Wnt11 during epaxial myotome morphogenesis.

## Discussion

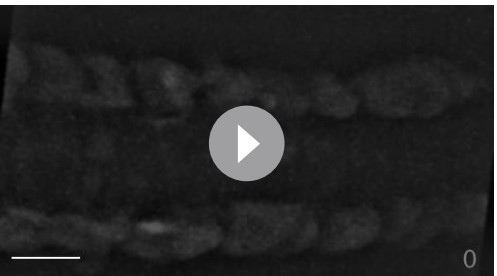

**Video 5.** Onset of dorsal somite extension in *wnt11* morphant embryo. *wnt11* was knocked down using the PhotoMO approach (Figure 7A) in a Tg(*zic1:GFP*) embryo. Dorsal view of time-lapse in vivo imaging of 24 ss Tg(*zic1:GFP*) embryo. 15[th] somite is positioned in the center, z-stacks were imaged every 10 min, time is displayed in min. Anterior to the left. Scale bar = 50 µm.

https://elifesciences.org/articles/71845/figures#video5

Here, we elucidated the key developmental process underlying the epaxial myotome morphogenesis to cover the neural tube in the teleost, medaka. DM cells have been known to serve as a progenitor pool for myotomal and dermal cells (*Ben-Yair and Kalcheim, 2005*; *Hollway et al., 2007*; *Stellabotte and Devoto, 2007*). In our study, we showed that DM cells at the tip of the dorsal somite form unique large protrusions, guiding the myotome towards the top of the neural tube. Furthermore, these DM cells do not give rise to myotomal cells during embryonic development. We found that *zic1* and its direct downstream target *wnt11* are crucial factors for this morphogenetic process. We thus revealed a novel role of non-myogenic DM cells during epaxial myotome morphogenesis. Furthermore, to our best knowledge, this work demonstrates for the first time that the neural tube coverage by

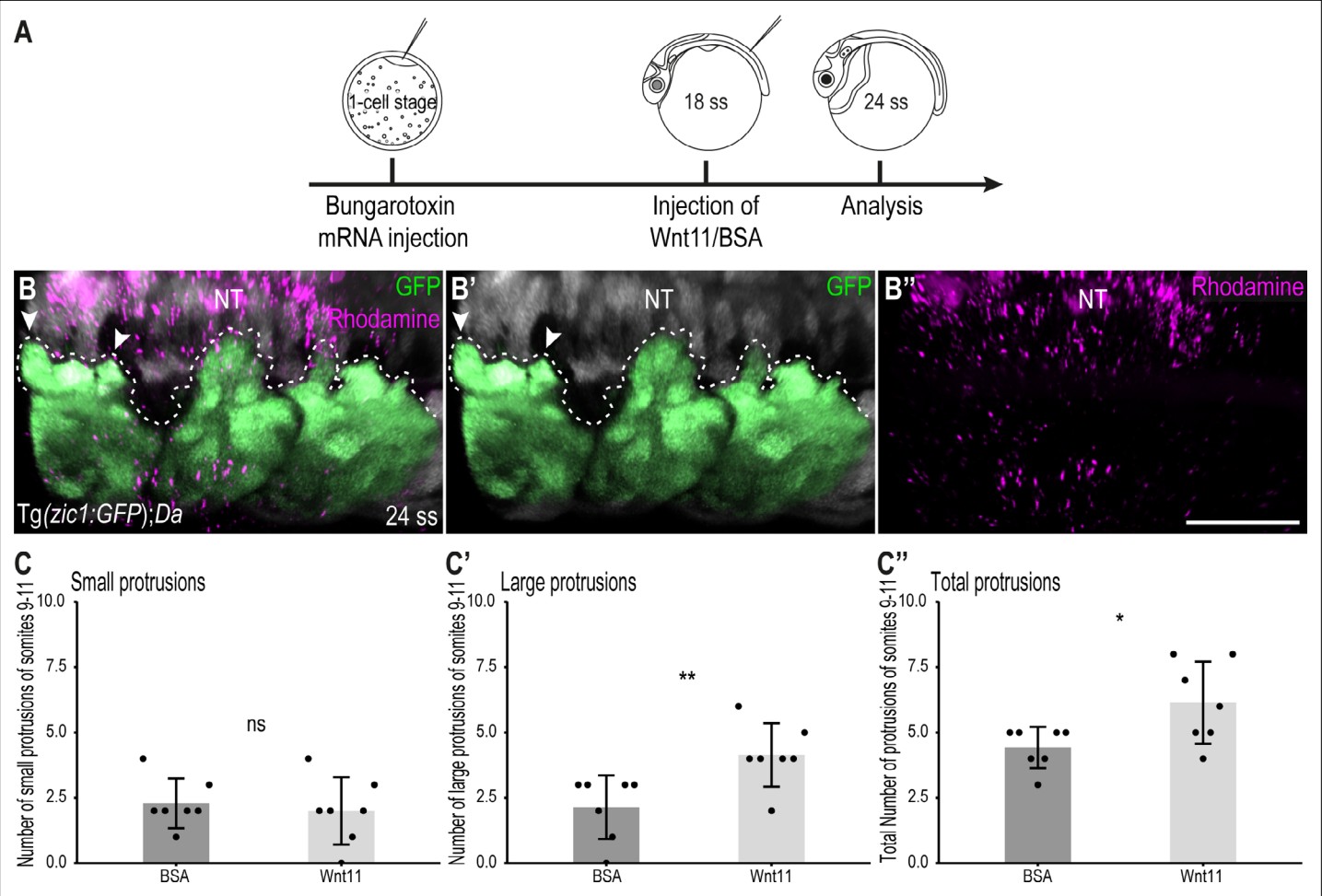

**Figure 8.** Wnt11 injections partially rescue the Tg(*zic1:GFP*);*Da* protrusion phenotype. (**A**) Schematic representation of Wnt11 protein injection on top of the 10th somite of 18 ss Tg(*zic1:GFP*);*Da* embryos. (**B-B″**) Dorsal view of dorsal somites of Tg(*zic1:GFP*);*Da* embryos injected with Wnt11-Dextran Rhodamine mix (magenta). GFP signals in dorsal somites are highlighted in green. Protrusions are outlined. (**C-C″**) Quantification of dorsal protrusions of the 9th – 11th somite (somite at injection site plus adjacent somites) of embryos injected with BSA (n = 7 embryos) or hrWnt11(n = 7 embryos) (mean ± SD, p = 0.65 for small protrusions, p = 0.0095 for large protrusions, p = 0.03 for total protrusions). Anterior to the left. NT, neural tube. Scale bar = 50 µm. *p < 0.05, **p < 0.01, ns, not significant.

myotomes is driven by active cell movement, rather than passively by muscle growth. Consistent with this, in the absence of Zic1 activity, the myotome is unable to cover the neural tube at the hatching stage, despite increased dorsal myotome growth (*Figures 1L and 2*), supporting the involvement of the active cellular mechanism in this process. The model we propose for dorsal somite extension is shown in *Figure 10*.

In the present study, we described the characteristic behavior of dorsal DM cells and propose their guiding role for the epaxial myotome moving towards the top of the neural tube. During dorsal somite extension, DM cells at the tip of the dorsal somites form large, motile protrusions, which contain multiple bundles of filopodia dynamically branching out from the tip of the protrusions. These long protrusions could be beneficial for invading the restricted open space between the neural tube and the ectoderm. Additionally, previous studies showed that in migrating mesenchymal cells, the formation of lamellipodia is associated with higher migratory speed, whereas filopodia play an exploratory role and are associated with high directionality (*Leithner et al., 2016*; *Innocenti, 2018*). Large protrusions of wild-type DM cells consist of a lamellipodia-like core and multiple filopodia, which could account for fast dorsal somite extension with high accuracy. The detailed structure and function of the large protrusions needs to be further investigated in the future studies.

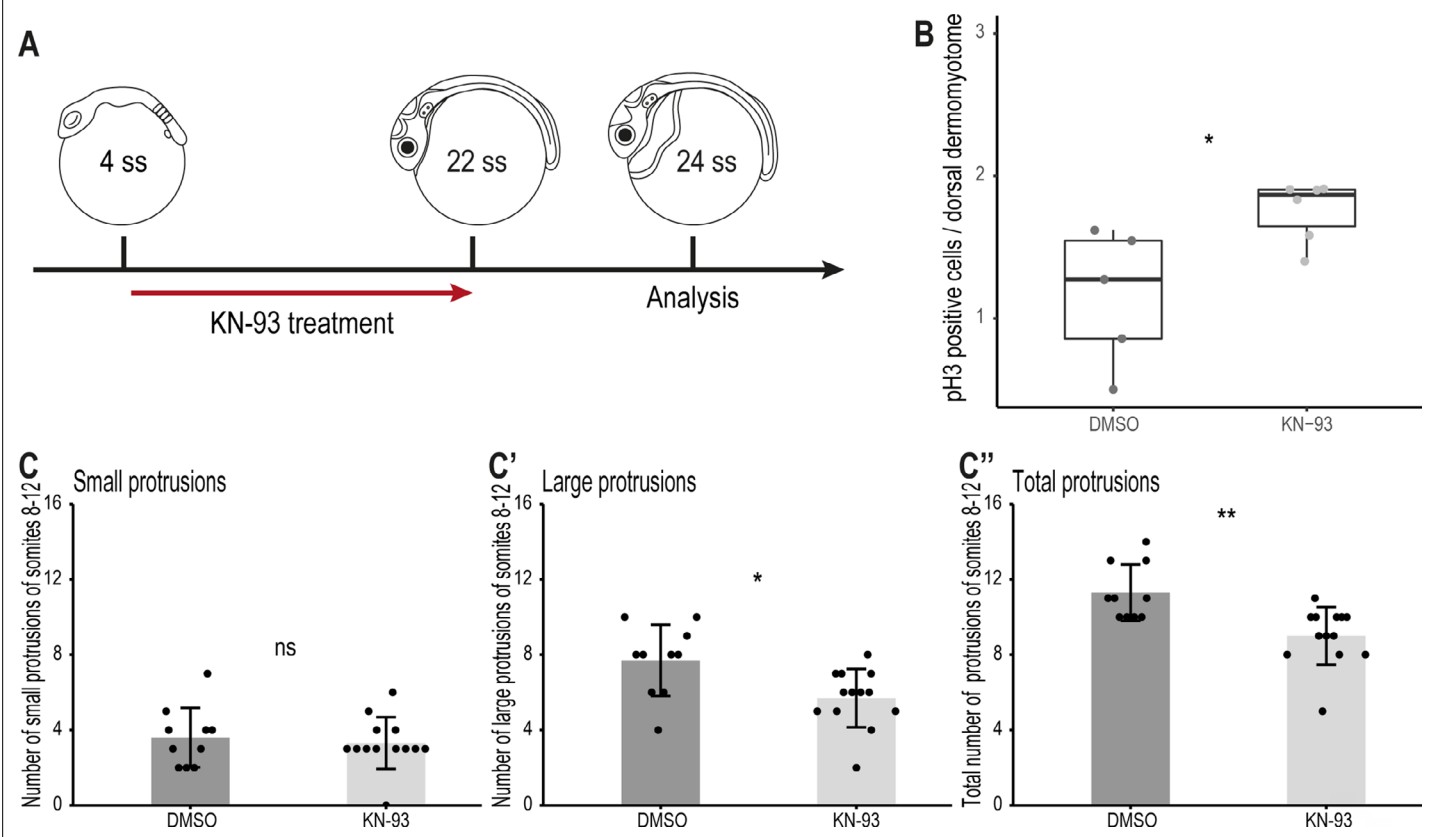

**Figure 9.** Wnt11 acts through the Wnt/Ca$^{2+}$ signaling pathway during dorsal somite extension. (**A**) Schematic representation of KN-93 treatment, embryos in the control group were treated with DMSO. (**B**) Embryos treated with KN-93 (n = 65.5 somites from 6 embryos) have significantly more pH3-positive cell in the dorsal DM compared to embryos of the control group (n = 52.5 somites from 5 embryos) (median, first and third quartiles, p = 0.045). (**C-C''**) Quantification of protrusions formed by the 8$^{th}$-12$^{th}$ somites of Tg(*zic1:GFP*) embryos treated with DMSO (n = 10 embryos) or KN-93 (n = 13 embryos) (mean ± SD, p = 0.65 for small protrusions, p = 0.014 for large protrusions, p = 0.0017 for total protrusions). *p < 0.05, **p < 0.01, ns, not significant.

In zebrafish, hypaxial muscle precursors delaminate from the ventral tip of the DM and migrate collectively in a cell stream to the prospective pectoral fin bud region while the cells at the leading edge form long filopodia (*Haines et al., 2004*; *Talbot et al., 2019*). Thus, DM cells, regardless of whether dorsal or ventral, seem to have the potential to form protrusions and to exhibit migratory behavior. However, regulation of migratory behavior could be different as the migration of hypaxial muscle precursors is mediated by the receptor tyrosine kinase Met (*Haines et al., 2004*) which is not expressed in zebrafish dorsal DM cells.

At later stages of dorsal somite extension, we observed that DM cells delaminate from the tip of the dorsal somites and progressively occupy the space between the opposing dorsal somites. These mesenchymal DM cells actively form protrusions towards neighboring mesenchymal DM cells and DM cells of the somites, thus forming a dense cellular network on top of the neural tube. This could provide a communication platform for the opposing somites to meet at the right position, exactly on top of the neural tube. Indeed, ablation of mesenchymal DM cells resulted in transient opening of the gap between the somites, suggesting that these cells at least exert a tension to hold the positions of the dorsal tips of the left and right somite together during neural tube coverage. Interestingly, our lineage tracing experiment revealed that the dorsal DM cells at the tip of somites differentiated into cells of non-myotomal lineages including mural cells of blood vessels and mesenchymal cells in the dorsal fin fold. These observations are consistent with previous studies reporting their somitic origins in zebrafish and *Xenopus* (*Garriock and Krieg, 2007*; *Lee et al., 2013*; *Ando et al., 2016*). However, our data also showed that these DM cells do not appear to contribute to the myotome at the hatching

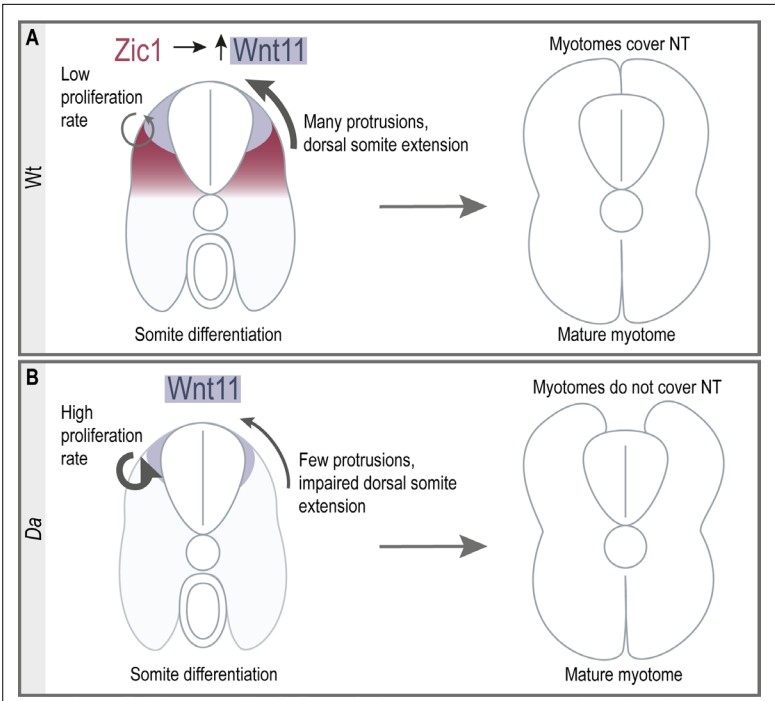

**Figure 10.** Summary of dorsal somite extension in Wt and the *Da* mutant. (**A**) Zic1 induces or maintains the expression of *wnt11* during somite differentiation. This leads to a reduced proliferative activity of the dorsal DM and increases protrusion formation of the dorsal DM cells which are non-myogenic. Ultimately, the non-myogenic DM guides the epaxial myotomes dorsally where the myotomes form a gapless muscle layer covering the neural tube at the end of embryonic development. (**B**) In the *Da* mutant *wnt11* expression is reduced in the dorsal somites. The dorsal DM cells show a high proliferative activity and reduced ability to form numerous large protrusions causing insufficient dorsal somite extension. This results in the incomplete coverage of the neural tube by the epaxial myotomes at the end of embryonic development.

stage. We thus propose that the neural tube coverage is driven by a specific population of dorsal DM cells, which do not contribute to myotomal cells during embryonic development.

The ability of the motile DM cells to guide the underlying myotome dorsally assures complete coverage of the neural tube by the epaxial myotomes. Considering the fact that DM cells are in close cell-cell contact with the underlying myotome, it is plausible that they are biomechanically coupled with myotomal cells, facilitating dorsal myotome extension. A similar mechanism has been investigated in mouse neural tube formation. During mouse neural tube closure, the surface ectoderm, overlying the neuroepithelium, forms cell protrusions towards the midline. Disruption of membrane ruffles, a form of lamellipodia, in the surface ectoderm results in incomplete fusion of the subjacent neuroepithelium (*Rolo et al., 2016*).

Our data suggest that in medaka dorsal somites Wnt11 exerts its effect through promotion of protrusion formation and down-regulation of cell proliferation in the dorsal DM. Regarding the protrusion formation, previous studies of migrating mesendodermal cells in the zebrafish mutant *silberblick* and neural crest cells in *Xenopus* embryos consistently reported that Wnt11 is involved in the oriented elongation and stabilization of protrusions (*Ulrich et al., 2003*; *De Calisto et al., 2005*; *Matthews et al., 2008*). Similarly, myocardial cells in the heart of *Wnt11* mutant mice also form fewer protrusions (*Zhou et al., 2007*). Thus, regulation of cell protrusions is a conserved function of Wnt11, observed in various developmental processes. However, the relationship between Wnt11 and cell proliferation could be context-dependent, as it negatively regulates proliferation in mouse neonatal hearts (*Touma, 2017*), while it promotes cell proliferation in mouse intestinal epithelial cell culture (*Ouko et al., 2004*). Hence, the function of Wnt11 in medaka somites is unique in that it promotes tissue elongation by regulating a balance between proliferative and migrative activity.

By inhibiting the Wnt/Ca$^{2+}$ signaling pathway we showed that Wnt11 probably acts through this non-canonical Wnt signaling pathway during dorsal somite extension. Likewise, previous studies in

*Xenopus* have shown that Wnt11 acts through this pathway during cell migration into the dorsal fin fold. There it regulates epithelial–mesenchymal transition in a distinct dorsal somite cell population which, together with a population of neural crest cells, contribute to the mesenchyme of the dorsal fin fold (*Garriock and Krieg, 2007*). Furthermore, during convergent extension in vertebrate gastrulation, Wnt/Ca2+ signaling pathway can regulate cell adhesion by promoting actin polymerization (*Choi and Han, 2002*; *Kohn and Moon, 2005*). Whether Wnt11 functions cell-autonomously or not is context dependent (*De Calisto et al., 2005*; *Garriock and Krieg, 2007*; *Gros et al., 2009*). In our study, administration of exogenous Wnt11 protein rescued the protrusive behavior of the surrounding DM cells, suggesting that Wnt11 can function cell non-autonomously during dorsal somite extension.

Importantly, our RNA-seq and ChIPmentation analyses revealed that Zic1 has diverse downstream target genes including various developmental genes. This suggests that the dorsal myotome is established via pleiotropic actions of Zic1; Wnt11 may not be a sole factor for dorsal somite extension, although it was suggested to be crucial in the present study. The semaphorin-plexin pathway may play a role, since previous studies suggested, besides its implication in axon guidance and neural cell migration, a role in non-neural cell migration (*Alto and Terman, 2017*). Furthermore, cell migration and tissue deformation are often linked with the extracellular matrix (ECM), which provides guiding or restraining cues influencing cell movements. Previous studies showed that during mouse epaxial myotome development, ECM composition changes dynamically, which is tightly accompanied by epaxial muscle morphogenesis; while the laminin content in the ECM decreases, increasing tenascin and stable fibronectin contents potentially promotes the alignment of myofibers and their final organization (*Deries et al., 2012*). Furthermore, recent studies suggested that cells can remodel the surrounding ECM and thereby increase their own motility. For example, during zebrafish gastrulation, the metalloproteinase *mmp14* is expressed by migrating endoderm cells and degrades laminin and fibronectin, components of the ECM (*Hu et al., 2018*). In this context, of particular importance is our identification of *Adamts20*, encoding a proteoglycanase, as a dorsal-high Zic1-target gene in the somites (*Supplementary file 1*). Since Adamts20 is known to play a pivotal role in embryonic melanoblast migration by remodeling the dermal ECM (*Rao et al., 2003*; *Silver et al., 2008*), Adamnts20 could thus facilitate migration of Wnt11-expressing DM cells through remodeling of the ECM. Further functional analysis of dorsal-high Zic1-target genes identified in somites will provide useful insight into the molecular network driving dorsal somite extension.

Finally, in vertebrates and especially in fish, body shape and muscle morphology are closely linked, since a majority of the body mass consist of muscular tissue. Previous studies in fish populations have shown that speciation and adaptation to a specific aquatic habitat are associated with a changes in body depth, a measurement of the trunk dorsoventral axis (*Tobler, 2008*; *Elmer et al., 2010*; *Weese et al., 2012*; *Fruciano, 2016*). In this context, the external appearance of the adult *Da* mutant is intriguing, in that it exhibits a teardrop body shape (typical for fish swimming in the middle layer like tuna), instead of a dorsally flattened one (typical for surface swimming fish like medaka). Since our study shows that the activity of *wnt11* could influence the body shape by regulating cell proliferation and the behavior of muscle progenitor cells, Wnt11 could be one of the crucial factors in evolution and diversity of body shape in fish. Furthermore, the expression of *zic1* (*Ohtsuka et al., 2004*; *Sun Rhodes and Merzdorf, 2006*; *Houtmeyers et al., 2013*) and *wnt11* (*Marcelle et al., 1997*; *Olivera-Martinez et al., 2004*; *Garriock et al., 2005*; *Matsui et al., 2005*) are strongly conserved among vertebrates, and we hypothesize that Wnt11-mediated morphogenesis of the somites represents an evolutionarily conserved mechanism that acts across vertebrates.

## Materials and methods

### Key resources table

| Reagent type (species) or resource | Designation | Source or reference | Identifiers | Additional information |
|---|---|---|---|---|
| Strain, strain background (*Oryzias latipes*) | d-rR | *Yamamoto, 1975* | | Medaka Southern wild type population |

*Continued on next page*

*Continued*

| Reagent type (species) or resource | Designation | Source or reference | Identifiers | Additional information |
|---|---|---|---|---|
| Strain, strain background (*O. latipes*) | *Double anal fin* (*Da*) mutant | **Ohtsuka et al., 2004** | | |
| Strain, strain background (*O. latipes*) | Tg(*zic1:GFP,zic4:DsRed*) | **Kawanishi et al., 2013** | | |
| Strain, strain background (*O. latipes*) | Tg(*zic1:zic1-Myc,zic4:DsRed*);*Da* | this paper | | BAC construct *zic1:zic1-Myc/zic4:DsRed* was introduced into the *Da* mutant |
| Antibody | Anti-Digoxigenin-AP Fab fragments, peroxidase-conjugated (sheep polyclonal) | Roche (Germany) | 11093274910 | IHC (1:2000) |
| Antibody | Anti-GFP (rabbit polyclonal) | Clontech (Mountain View, California, USA) | 632592, RRID: 2336883 | IHC (1:500) |
| Antibody | Anti-mouse Alexa 555 (goat polyclonal) | Thermo Fisher (Waltham, Massachusetts, USA) | A-21422, RRID: AB_2535844 | IHC (1:500) |
| Antibody | Anti-Pax3/7 (DP312) (mouse monoclonal) | **Davis et al., 2001** | | IHC (1:100) |
| Antibody | Anti-PCNA (rabbit polyclonal) | Abcam (Cambridge) | ab18197, RRID:AB_444313 | IHC (1:200) |
| Antibody | Anti-pH3 (Ser10) (rabbit polyclonal) | Milipore (Burlington, Massachusetts, USA) | 06–570,RRID: AB_310177 | IHC (1:200) |
| Antibody | Anti-rabbit Alexa 488 (donkey polyclonal) | Life Technologies (Carlsbad, California, USA) | A-11008, RRID: AB_143165 | IHC (1:500) |
| Recombinant DNA reagent | gRNA expression vector (DR274) | Addgene (Watertown, Massachusetts, USA) | 42,250 | T7 promoter for in vivo transcription |
| Recombinant DNA reagent | pCRIITOPO-*wnt11* | **Kawanishi et al., 2013** | For ISH at 12 ss | T7 promoter for RNA generation |
| Recombinant DNA reagent | pMTB-AC-TagGFP2 | this paper | | T7 promoter for mRNA generation |
| Recombinant DNA reagent | pGEM-Teasy(Wnt11_ISH) | this paper | For ISH at 22 ss | Sp6 promoter for RNA generation |
| Recombinant DNA reagent | pMTB-memCherry | **Xiong et al., 2014** | | T7 promoter for mRNA generation |
| Recombinant DNA reagent | pMTB-memb-mTagBFP2 | **Collins et al., 2018** | | Vector with Tol2 sites |
| Recombinant DNA reagent | pMTB-t7-alpha-bungarotoxin J4 lab stock | Addgene (Watertown, Massachusetts, USA) | 69,542 | T7 promoter for mRNA generation |
| Recombinant DNA reagent | pMTB-KikGR | This paper | | Sp6 promoter for mRNA generation |
| Recombinant DNA reagent | pSPORT6.1(*zic1*-ISH) | This paper | | T7 promoter for RNA generation |
| Recombinant DNA reagent | *zic1:GFP/zic4:DsRed* (BAC) | **Kawanishi et al., 2013** | | Vector with I-SceI meganuclease sites |
| Recombinant DNA reagent | *zic1:zic1-Myc/zic4:DsRed* (BAC) | This paper | | Vector with I-SceI meganuclease sites |

*Continued on next page*

*Continued*

| Reagent type (species) or resource | Designation | Source or reference | Identifiers | Additional information |
|---|---|---|---|---|
| Sequence-based reagent | PCR forward primer to amplify sgRNA template for in vivo transcription | *Lee et al., 2020* | | AAAAGCACCGACTCGGTG |
| Sequence-based reagent | PCR reverse primer to amplify sgRNA template for in vivo transcription | This paper | | GGTCAGGTATGATTTAAATGGTCAGT |
| Sequence-based reagent | PCR forward primer to generate *wnt11* ISH probe for 12 ss | *Kawanishi et al., 2013* | | CAAATGGCTAACACTGTCTCAAAC |
| Sequence-based reagent | PCR reverse primer to generate *wnt11* ISH probe for 12 ss | *Kawanishi et al., 2013* | | CTATTTGCAAACGTATCTCTCCAC |
| Sequence-based reagent | PCR forward primer to generate *wnt11* ISH probe for 22 ss | This paper | | CATGAAGAGCCGCTCTCACA |
| Sequence-based reagent | PCR reverse primer to generate *wnt11* ISH probe for 22 ss | This paper | | TCCCTGAGGTCTTGGAGTCC |
| Sequence-based reagent | RT-PCR forward primer for *gapdh* | This paper | | TGGCCATCAATGACCCGTTC |
| Sequence-based reagent | RT-PCR reverse primer for *gapdh* | This paper | | TAGTTTGCCTCCCTCAGCCT |
| Sequence-based reagent | RT-PCR forward primer for *wnt11* | This paper | | CGCTGGAACTGCTCCTCTAT |
| Sequence-based reagent | RT-PCR reverse primer for *wnt11* | This paper | | GCAGCGGACAAGGCATAAAC |
| Sequence-based reagent | RT-PCR forward primer for *zic1* | *Nakamura et al., 2014* | | AGCCCTTTCCGTGTCCGTTCC |
| Sequence-based reagent | RT-PCR reverse primer for *zic1* | *Nakamura et al., 2014* | | CCGACGTGTGGACGTGCATGT |
| Sequence-based reagent | Standard control | Gene Tools (Philomath, Oregon, US) | Morpholino | CCTCTTACCTCAGTTACAATTTATA |
| Sequence-based reagent | *wnt11* MO | This paper | Morpholino | CTTCATGATGGATGGAGGCTCCGGT |
| Sequence-based reagent | *wnt11* PhotoMO | This paper | Morpholino | CGGAGCCTPCATCCATCATG |
| Sequence-based reagent | Oligo annealing *wnt11_1* sgRNA template, forward | This paper | | TAGGTTCTGTCGGGAGACTTTA |
| Sequence-based reagent | Oligo annealing *wnt11_1* sgRNA template, reverse | This paper | | AAACTAAAGTCTCCCGACAGAA |
| Sequence-based reagent | Oligo annealing *wnt11_2* sgRNA template, forward | This paper | | TAGGCTGGATGTTTAACGGAG |
| Sequence-based reagent | Oligo annealing *wnt11_2* sgRNA template, reverse | This paper | | AAACCTCCGTTAAACATCCAG |
| Peptide, recombinant protein | BsaI-HF | New England Biolabs (Ipswich, USA) | R3733 | |
| Peptide, recombinant protein | EnGen Spy Cas9 NLS | New England Biolabs (Ipswich, USA) | M0646T | |
| Peptide, recombinant protein | Pancreatin | Wako (Japan) | 163–00142 | |

*Continued on next page*

*Continued*

| Reagent type (species) or resource | Designation | Source or reference | Identifiers | Additional information |
|---|---|---|---|---|
| Peptide, recombinant protein | Recombinant Human Wnt11 protein (hrWnt11) | R&D systems (Minneapolis, Minnesota, USA) | 6179-WN-010/CF | |
| Peptide, recombinant protein | Trypsin | Nacalai Tesque (Japan) | 35555–54 | |
| Chemical compound, drug | 4-nitro blue tetrazolium chloride (NBT) | Roche (Germany) | 11585029001 | |
| Chemical compound, drug | 5-bromo-4-chloro-3-indolyl phosphate (BCIP) | Roche (Germany) | 11383221001 | |
| Chemical compound, drug | Isogen | Nippon Gene (Japan) | 315–02504 | |
| Chemical compound, drug | KN-93 | Wako (Japan) | 114–00731 | |
| Chemical compound, drug | ML141 | Sigma-Aldrich (St. Louis, Missouri, USA) | SML0407 | |
| Chemical compound, drug | DAPI | Sigma-Aldrich (St. Louis, Missouri, USA) | D9542 | |
| Chemical compound, drug | Mayer's Hematoxylin Solution | Wako (Japan) | 131–09665 | |
| Chemical compound, drug | Propidium iodine | Life Technologies (Carlsbad, California, USA) | P3566 | |
| Chemical compound, drug | Rhodamine Phalloidin | Life Technologies (Carlsbad, California, USA) | R415 | |
| Commercial assay or kit | HiScribe T7 Quick High Yield RNA Synthesis Kit | New England Biolabs (Ipswich, USA) | E2050S | |
| Commercial assay or kit | KAPA HiFi HotStart ReadyMix | KAPA Biosystems (Wilmington, Massachusetts, USA) | KK2601 | |
| Commercial assay or kit | KAPA Stranded mRNA-seq Kit | KAPA Biosystems (Wilmington, Massachusetts, USA) | KK8420 | |
| Commercial assay or kit | MinElute Kit | Qiagen (Germany) | 28,004 | |
| Commercial assay or kit | Nextera Index Kit | Illumina (San Diego, California, USA) | FC-121–1011 | |
| Commercial assay or kit | Nextera Sample Preparation Kit | Illumina (San Diego, California, USA) | FC-121–1030 | |
| Commercial assay or kit | RNeasy Mini Kit | Qiagen (Germany) | 74,104 | |
| Commercial assay or kit | Super Script III Kit | Invitrogen (Waltham, Massachusetts, USA) | 18080051 | |
| Commercial assay or kit | THUNDERBIRD SYBR qPCR Mix | Toyobo (Japan) | QPS-201 | |
| Software, algorithm | Adobe Illustrator | Adobe (Mountain View, California, USA) | | Version 25.2.2 |
| Software, algorithm | Burrows-Wheeler Aligner (BWA) mapping software | *Li and Durbin, 2009* | | |

*Continued on next page*

*Continued*

| Reagent type (species) or resource | Designation | Source or reference | Identifiers | Additional information |
|---|---|---|---|---|
| Software, algorithm | FluoRender | *Wan et al., 2009* | | Version 2.19 |
| Software, algorithm | HOMER | *Heinz et al., 2010* | | |
| Software, algorithm | Imagej/ Fiji | *Schindelin et al., 2012* | | |
| Software, algorithm | MACS2 | *Zhang et al., 2008* | | Version 2.1.1.20160309 |
| Software, algorithm | STAR | *Dobin et al., 2013* | | |
| Software, algorithm | Trimmomatic | *Bolger et al., 2014* | | Version 0.32 |

## Animals and transgenic lines

Fish were raised and maintained under standard conditions. All experimental procedures and animal care were performed according to the animal ethics committee of the University of Tokyo. Sex was randomly assigned to experimental groups. Medaka d-rR stain was used as wild type, the *Da* mutant used in this study was previously described (*Ohtsuka et al., 2004*). The pre-existing transgenic line Tg(*zic1:GFP,zic4:DsRed*) (*Kawanishi et al., 2013*) was used, and the transgenic line Tg(*zic1:G-FP,zic4:DsRed*);*Da* was created by crossing *Da* mutants with Tg(*zic1:GFP,zic4:DsRed*). The transgenic line Tg(*zic1:zic1-Myc,zic4:DsRed*);*Da* was generated by modifying the BAC used to generate Tg(*zic1:GFP,zic4:DsRed*) (*Kawanishi et al., 2013*), by replacing the ORF of *GFP* with the ORF of *zic1* containing a sequence for a Myc-tag fused to its C-terminus. To establish the transgenic line, the BAC(*zic1:zic1-Myc,zic4:DsRed*) was co-injected with I-SceI Meganuclease (NEB) into one-cell stage *Da* embryos, as previously described (*Thermes, 2002*).

## Visualization of actin skeleton of protrusions

The AC-TagGFP2 sequence from the Actin-Chromobody plasmid (TagGFP2) (Chromotek) was cloned into the pMTB vector for mRNA generation. To investigate the actin skeleton of protrusions, cells of Wt and *Da* embryos were mosaically labeled using *Actin-Chromobody-GFP* (*AC-GFP*) mRNA. Embryos were injected at one-cell stage with 152 ng/μl *membrane-mCherry* mRNA. At four-cell stage, one cell was injected with 184 ng/μl *AC-GFP* mRNA. Embryos were raised until the desired stage and in vivo imaging was performed.

## Lineage tracing of mesenchymal DM cells

Mosaic labeling of cells was achieved by co-injecting 20 ng/μl pMTB-memb-mTagBFP2 plasmid with Tol2 mRNA into one-cell stage embryos of the transgenic line Tg(*zic1:GFP,zic4:DsRed*). Embryos were raised until 5 dpf and labeled cells were continuously observed until 8 dpf. Observation of cells was performed according to "In vivo imaging and in vivo time-lapse imaging".

## Photoconversion of Dorsal DM Cells

Wild type embryos were injected with 200 ng/μl *kikGR* mRNA at one-cell stage. Single dorsal DM cells exhibiting protrusions at the dorsal tip of 10th and 20th somites at 3.5 dpf (st. 29) were selectively photoconverted with a 405 nm laser under an LSM 710 confocal microscope and imaged again at 9 dpf (hatching stage). During the experiment, 8 embryos were labeled at both 10th and 20th somites, and 1 embryo was labeled only at the 10th somite.

### Morpholino injection

*wnt11* MO was injected at one-cell stage. We used 12.5 μM *wnt11* MO after finding that injection of 25 μM *wnt11* MO gives rise to gastrulation delay, leading to smaller head and trunk phenotypes at the onset of dorsal somite extension. Injection of 12.5 μM *wnt11* MO mostly did not hamper gastrulation. We excluded any MO-injected embryos showing gastrulation defects in the analyses at later timepoints.

## Photo-Morpholino injection

*wnt11* Sense-Photo-Morpholino (PhotoMO) and *wnt11* antisense Morpholino were annealed in a ratio 2:1. Microinjection of 25 µM of the annealed oligonucleotides was performed at one-cell stage, and embryos were raised until 4 ss in the dark. Photo-cleavage was performed using the 10 x objective and the DAPI filter of a Keyence BZ-9000 Biorevo microscope (Keyence). Embryos were mounted, dorsal side facing up, in 1% Methylcellulose in a glass bottom dish (Wako) and illuminated for 30 min. After Photo-cleavage, embryos were dechorionated and raised until the desired stage for subsequent analysis.

## Generation of sgRNAs

The online tool CCTop (*Stemmer et al., 2015*) was used to design sgRNAs. sgRNAs were generated as previously described by *Hwang et al., 2013*. Oligonucleotides were annealed and ligated into DR274 vector (Addgene), which was previously linearized using BsaI-HF (New England Biolabs). Template for in vivo transcription of sgRNA was amplified by PCR. In vivo transcription was performed using the HiScribe T7 Quick High Yield RNA Synthesis Kit (New England Biolabs) and RNA was purified using the RNeasy Mini kit (Qiagen). Quantity and quality of sgRNA were analyzed using a Nano-Drop and agarose gel electrophoresis. sgRNA cleaving activity was confirmed by an in vivo assay.

| sgRNA description | Target site [PAM] |
| --- | --- |
| *wnt11_1* | TTCTGTCGGGAGACTTTA[TGG] |
| *wnt11_2* | GCTGGATGTTTAACGGAG[TGG] |

### *wnt11* sgRNA injection

Microinjections were performed using 25 ng/µl *wnt11_1* sgRNA, 25 ng/µl *wnt11_2* sgRNA and 100 ng/µl Cas9-NLS (New England Biolabs) at one-cell stage. To analyze phenotypes, embryos were mounted in 1% methylcellulose and imaged using a Leica M165 FC stereo microscope.

## Injection of human recombinant Wnt11 protein into *Da* mutant somite

To immobilize embryos, embryos from the Tg(*zic1:GFP,zic4:DsRed*);*Da* transgenic line were injected with 25 ng/µl *α-bungarotoxin* mRNA at one-cell stage. At 18 ss embryos were mounted in 1% low melting agarose in 1 x Yamamoto's Ringer Solution and oriented with the dorsal side facing upwards. Embryos were injected on top of the 10[th] somite with a mix containing Dextran Rhodamine (Thermo Fisher) and 1.7 ng hrWnt11 protein (R&D Systems) or BSA (Sigma-Aldrich) and raised to 24 ss, followed by in vivo imaging and analysis.

## Laser ablation of mesenchymal DM cells

Tg(*zic1:GFP,zic4:DsRed*) embryos were immobilized by injecting 25 ng/µl *α-bungarotoxin* mRNA at one-cell stage and raised until 5.5 dpf. Z-stacks of the 9[th] (control somite) and 10[th] somite were imaged before ablation using the 40 x objective of a Zeiss LSM 710 confocal microscope. Laser ablation protocol was adapted from *Morsch et al., 2017* and *Volpe et al., 2020*. Prior to laser ablation the 40 x objective and additional the 2.2 x zoom were used to focus only on the ablation site, the mesenchymal DM cells between the 10[th] somites. The ablation site was chosen to prevent damage of dorsal somites and the neural tube. Laser ablation was perforemd using a 405 nm laser at 80% power for 10 min on 'continous' mode. The 9[th] and 10[th] somite were subject to in vivo imaging directly after laser ablation and every 8 h until 40 hpa.

## KN-93 treatment

Dechorionated 4 ss embryos were treated with 30 µM KN-93 (Wako) or DMSO (Sigma-Aldrich) in 1 x Yamamoto's Ringer Solution at 28 °C, in the dark until 22 ss.

## ML141 treatment

Embryos at 4 ss were dechorionated and treated with 500 µM ML141 (Sigma-Aldrich) or DMSO in 1 x Yamamoto's Ringer Solution at 28 °C, in the dark until 24 ss.

## In vivo imaging and in vivo time-lapse Imaging

To immobilize embryos for in vivo imaging, embryos were injected at one-cell stage with 25 ng/µl α-bungarotoxin mRNA (*Swinburne et al., 2015*; *Lischik et al., 2019*). Embryos were mounted in 1% Low melting agarose (Sigma-Aldrich) in 1 x Yamamoto's Ringer Solution in a glass-bottomed petri dish (IWAKI) and oriented with the dorsal side facing down. Imaging was performed using a Zeiss LSM 710 confocal microscope system (Zeiss) equipped with an inverted stand and a Zeiss AXIO Observer Z1 and a T-PMT detector. The embryos were positioned with the 5th or 10th somite in the center and images were acquired using a 40 x water objective. For the in vivo time-lapse imaging, the 10th or 15th somite was positioned in the center, z-stacks were imaged in a 600 s interval for 10–15 hr. Image analysis was performed in Fiji using the "Image Stabilizer" Plugin, the FFT Bandpass filter and the "Draw_arrows" Plugin to draw customized arrows (*Li, 2008*; *Daetwyler et al., 2020*).

## Whole mount in situ hybridization

Whole mount in situ hybridization was performed as previously described with the following modifications (*Takashima et al., 2007*). Embryos were fixed in 4% PFA/1.5 x PTW at 4 °C, overnight. Hybridization was performed at 65 °C, overnight. Samples were treated with alkaline-phosphatide anti-DIG-AP Fab fragments (1:2000, Roche). Signals were developed using 4-nitro blue tetrazolium chloride (NBT, Roche) and 5-bromo-4-chloro-3-indolyl phosphate (BCIP, Roche).

## Whole-mount immunohistochemistry

Embryos were fixed in 4% PFA/PBS for 2 hr at room temperature or at 4 °C overnight. Samples were permeabilized with 0.5% TritonX-100 (Wako) in 1 x PBS for 1–2 hr and blocked in blocking solution (2% BSA (Sigma-Aldrich), 1% DMSO, 0.2% TritonX-100 in 1 x PBS) for 2–4 hr at room temperature. Samples were incubated with respective primary antibody diluted in blocking solution at 4 °C, overnight. After an additional 4 hr blocking step, samples were incubated with respective secondary antibodies diluted in blocking solution, at 4 °C, overnight. Samples were stored in 1 x PBS at 4 °C until imaging.

## Vibratome sectioning

Samples were mounted in 4% agarose in 1 x PBS. 40 µm or 200 µm sections were obtained by a Vibratome (Leica, Vibratome). The sections were mounted on a glass slide (Matsunami) in 60% glycerol (Merck, Wako) and stored at 4 °C until imaging. Images were acquired using the 40 x water objective of a Zeiss LSM 710 confocal microscope.

## Histological sections

Dechorionated embryos were fixed in Bouin's solution overnight, followed by a gradual dehydration using ethanol. Samples were embedded in Technovit 7,100 (Heraeus Kulzer) and sectioned into 5–6 µm thick sections. Sections were stained with hematoxylin (Wako) and imaged using the 1.6 x objective of a Leica M165 FC fluorescent stereo microscope.

## Image processing and statistical analysis

Image processing was performed with the image processing software Fiji. The 3D-recreation of in vivo imaging date was created using FluoRender (*Wan et al., 2009*). Measurements of morphological features (distance between myotome tips, area of cross-section of dorsal somites, diameter of myofibers, distance between dorsal somite tip and the tip of neural tube, somite height) was performed by averaging the analysis of the feature from three consecutive Y planes. RStudio was used for the statistical analysis and representation of the data. In bar plots mean and error limits, defined by the standard deviation, are indicated. In box plots median first and third quantiles are indicated. Statistical significance was determined by un-paired t-tests, a p-value $p < 0.05$ was considered as significant. In the figure legends sample size (n) and number of individuals used in the experiment are stated. Sample sizes were not predetermined using statistical methods, but the sample sizes used are similar to those generally used in the field. To compare experimental groups, the allocation was performed randomly, without blinding.

## RT-PCR of cDNA generated from embryonic tails

To investigate the gene expression in tails of embryos, tails were dissected anterior from the first somite. Ten tails were pooled together and RNA was isolated using Isogen (Nippon Gene). RNA was purified using the RNeasy Mini kit (Qiagen) and reverse transcribed to cDNA using the Super Script III Kit (Invitrogen). RT-PCR was performed using the Thunderbird Sybr qPCR Mix (Toyobo) following manufacturer's instructions and run in the Agilent Mx3000P qPCR System (Agilent). Normalization of relative quantities was performed against *gapdh* expression, followed by analysis with excel and RStudio.

## Isolation of dorsal and ventral somite cells for ATAC-Seq and RNA-Seq

Yolk and head region were removed from 22 ss Tg(*zic1:GFP,zic4:DsRed*) embryos, and remaining trunk-tail pieces were incubated with 10 mg/ml pancreatin (Wako) at room temperature for 5–10 min to loosen the adhesion between tissues while maintaining the integrity within each tissue. Epidermis and intestinal tissues were removed, and then somites were isolated from the neural tube. Collected somites were subsequently dissociated into individual cells in 0.5% (w/v) Trypsin (Nacalai Tesque) at 37 °C for 10 min, and the dissociation was stopped by adding the same volume of 15% (v/v) FBS / Leiboviz's L-15 (Life Technologies). Dissociated cells were washed with PBS, and sorted into GFP-positive (dorsal) and negative (ventral) cells using FACSAria III (BD Biosciences). Dead cells were detected by Propidium iodide (Life Technologies) and removed.

## RNA-Seq

Total RNA was extracted from sorted somite cells using RNeasy Mini kit (Qiagen). mRNA was enriched by poly-A capture and mRNA-seq libraries were generated using KAPA Stranded mRNA-seq Kit (KAPA Biosystems). Libraries were generated from two biological replicates, and sequenced using the Illumina HiSeq 1,500 platform.

## RNA-Seq data processing

The sequenced reads were pre-processed to remove low-quality bases and adapter derived sequences using Trimmomatic v0.32 (*Bolger et al., 2014*), and aligned to the medaka reference genome (HdrR, ASM223467v1) by STAR (*Dobin et al., 2013*). Reads with mapping quality (MAPQ) larger than or equal to 20 were used for the further analyses.

## ATAC-Seq

ATAC-seq was performed as previously described (*Buenrostro et al., 2013*) with some modifications. Approximately 4000 sorted somite cells were used for each experiment. After washing with PBS, cells were resuspended in 500 µl cold lysis buffer (10 mM Tris-HCl pH 7.4, 10 mM NaCl, 3 mM MgCl$_2$, 0.1% Igepal CA-630), centrifuged for 10 min at 500 g, supernatant was removed. Tagmentation reaction was performed as described previously (*Buenrostro et al., 2013*) with Nextera Sample Preparation Kit (Illumina). After tagmented DNA was purified using MinElute kit (Qiagen), two sequential PCRs were performed to enrich small DNA fragments. First, a 9-cycle PCR was performed using indexed primers from Nextera Index Kit (Illumina) and KAPA HiFi HotStart ReadyMix (KAPA Biosystems), amplified DNA was size selected for a size less than 500 bp using AMPure XP beads (Beckman Coulter). A second 7-cycle PCR was performed using the same primers as for the first PCR. PCR product was purified by AMPure XP beads. Libraries were generated from two biological replicates, and sequenced using the Illumina HiSeq 1,500 platform.

## ChIPmentation

Yolk and head region were removed from 22 ss Tg(*zic1:zic1-Myc,zic4:DsRed*);*Da* embryos, followed by an incubation with 10 mg/ml pancreatin (Wako) at room temperature for 5–10 min. Epidermis and intestinal tissues were removed and somites were isolated from the neural tube. ChIP was performed as previously described with the following modifications (*Nakamura et al., 2014*). Isolated somites were fixed with 1% formaldehyde for 8 min at room temperature then quenched by adding glycine (200 mM final) and incubating on ice for 5 min. After washing with PBS, cell pellets were stored at −80 °C. Approximately 1.8 × 10$^6$ cells were thawed on ice, suspended in lysis buffer (50 mM Tris-HCl (pH 8.0), 10 mM EDTA, 1% SDS, 20 mM Na-butyrate, complete protease inhibitors, 1 mM PMSF) and

sonicated 10 times using a Sonifier (Branson) at power 5. Chromatin lysates were collected by centrifugation and diluted 10-fold with RIPA ChIP buffer (10 mM Tris-HCl (pH 8.0), 140 mM NaCl,1 mM EDTA, 0.5 mM EGTA, 1% Triton X-100, 0.1% SDS, 0.2% sodium deoxycholate, 20 mM Na-butyrate, complete protease inhibitors, 1 mM PMSF) followed by an incubation with antibody/protein A Dynabeads (Invitrogen) complex at 4 °C, overnight, while rotating. Immunoprecipitated samples were washed three times with RIPA buffer (10 mM Tris-HCl (pH 8.0), 140 mM NaCl,1 mM EDTA, 0.5 mM EGTA, 1% Triton X-100, 0.1% SDS, 0.2% sodium deoxycholate) and once with TE buffer. After the washing steps 150 µl of Tris-HCl was added.

Library preparation for ChIPmentation was performed as previously described (*Schmidl et al., 2015*) with the following modifications. Twenty-four µl of Tagmentation reaction mix (10 mM Tris-HCl pH8.0, 5 mM MgCl$_2$, 10%(v/v) N,N-dimethyl formamide) and 1 µl of Tagment DNA Enzyme from Nextera Sample Preparation Kit (Illumina) were added to the DNA-beads complex and incubated for 70 s at 37 °C. A total of 150 µl ice-cold RIPA buffer was added and incubated for 5 min on ice. The DNA-beads complex was washed with RIPA buffer and TE buffer, suspended in 50 µl lysis buffer and 3 µl of 5 M NaCl, and incubated at 65 °C, overnight. The sample was incubated for 2 hr with 2 µl of 20 mg/ml ProteinasK (Roche), and subjected to AMPure XP beads (Beckman Coulter) purification. The library was amplified by 18-cycle PCR using indexed primers from Nextera Index Kit (Illumina) and KAPA HiFi HotStart ReadyMix (KAPA Biosystems). For the input chromatin, tagmentation reaction was performed after DNA purification. Libraries were generated from two biological replicates, and sequenced using the Illumina HiSeq 1,500 platform.

### ChIPmentation and ATAC-Seq data processing

The sequenced reads were pre-processed to remove low-quality bases and adapter derived sequences using Trimmomatic v0.32 (*Bolger et al., 2014*) and aligned to the medaka reference genome (HdrR, ASM223467v1) by BWA (*Li and Durbin, 2009*). Reads with mapping quality (MAPQ) larger than or equal to 20 were used for the further analyses. MACS2 (version 2.1.1.20160309) (*Zhang et al., 2008*) was used to call peaks and generate signals per million reads tracks using following options; ChIPmentation: -g 600000000 -B --SPMR `--keep-dup` 2, ATAC-seq: `--nomodel --extsize` 200 `--shift` –100 g 600000000 -q 0.01 -B `--SPMR`. The input chromatin was used as control of ChIPmentation.

For ChIPmentation, peak regions called by two biological replicates were used as reliable peaks.

### Motif analyses of Zic1 ChIPmentation peaks

Motifs enriched at reliable ChIPmentation peaks were analyzed by findMotifsGenome command of HOMER (*Heinz et al., 2010*) using default parameters.

### Identification of Zic1 target genes

Differentially expressed genes were identified using DESeq2 (padj <0.01). Each reliable ChIP peak was associated to the nearest TSS, and the gene was defined as Zic1-target gene if the distance between the peak and the TSS was closer than 50 kb.

### Gene ontology and pathway analyses

The gene ontology enrichment analyses and pathway enrichment analyses were performed using the Gene Ontology Resource (*Ashburner et al., 2000*; *Gene Ontology Consortium, 2021*; *Gene Ontology Consortium, 2021*).

## Acknowledgements

We thank the members of the Takeda laboratory for constructive feedback and discussions on the project. We are greatful for Y Yamagichi and M Funato for fish husbandry. This work was supported by Japan Society for the Promotion of Science (JSPS) KAKENHI Grant Numbers JP15H05859 (HT), JP19K23741 (TK) and JP18K14620 (RN) and Japan Science and Technology Agency CREST Grant Number JPMJCR13W3 (HT).

## Additional information

### Funding

| Funder | Grant reference number | Author |
|---|---|---|
| Japan Society for the Promotion of Science | JP15H05859 | Hiroyuki Takeda |
| Japan Society for the Promotion of Science | JP19K23741 | Toru Kawanishi |
| Japan Society for the Promotion of Science | JP18K14620 | Ryohei Nakamura |
| Japan Science and Technology Agency | JPMJCR13W3 | Hiroyuki Takeda |

The funders had no role in study design, data collection and interpretation, or the decision to submit the work for publication.

### Author contributions

Ann Kathrin Heilig, Conceptualization, Data curation, Formal analysis, Investigation, Methodology, Validation, Visualization, Writing - original draft, Writing - review and editing; Ryohei Nakamura, Data curation, Formal analysis, Funding acquisition, Investigation, Validation, Writing - review and editing; Atsuko Shimada, Data curation, Investigation, Writing - review and editing; Yuka Hashimoto, Data curation, Formal analysis, Investigation; Yuta Nakamura, Resources; Joachim Wittbrodt, Supervision, Writing - review and editing; Hiroyuki Takeda, Conceptualization, Funding acquisition, Supervision, Writing - review and editing; Toru Kawanishi, Conceptualization, Funding acquisition, Investigation, Resources, Supervision, Writing - review and editing

### Author ORCIDs

Joachim Wittbrodt (iD) http://orcid.org/0000-0001-8550-7377
Hiroyuki Takeda (iD) http://orcid.org/0000-0002-7932-6358
Toru Kawanishi (iD) http://orcid.org/0000-0001-7038-9769

### Ethics

All experimental procedures and animal care were performed according to the animal ethics committee of the University of Tokyo (Approval No. 20-02).

### Decision letter and Author response

Decision letter https://doi.org/10.7554/eLife.71845.sa1
Author response https://doi.org/10.7554/eLife.71845.sa2

## Additional files

### Supplementary files

• Supplementary file 1. Gene expression profiles and distances to nearest Zic1 ChIP peak.
• Supplementary file 2. Full list of GO terms enriched in dorsal-high Zic1 target genes.
• Supplementary file 3. Full list of GO terms enriched in dorsal-low Zic1 target genes.
• Transparent reporting form

### Data availability

Sequencing data generated in this study have been submitted to the DDBJ BioProject database under accession number PRJDB11712.

The following dataset was generated:

| Author(s) | Year | Dataset title | Dataset URL | Database and Identifier |
|---|---|---|---|---|
| Helig AK, Nakamura R, Hashimoto Y | 2022 | The function of transcription factor Zic1 in dorsalization of the medaka somite | https://ddbj.nig.ac.jp/resource/bioproject/PRJDB11712 | DDBJ BioProject, PRJDB11712 |

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
