## [Editor Report]

This study addresses an interesting and underexplored question in developmental biology, specifically cell migration and muscle development. It builds upon prior analysis of the medaka *Double anal fin* (*Da*) mutants by using detailed bioinformatic and time-lapse analysis to explain dorsal somite extension. The authors show that dorsal muscle morphogenesis is actively guided by dorsal dermomyotome cells, rather than being passively shaped by physical constraints alone. Looking downstream of *Da*, they show that Wnt signaling is central to dorsal extension of the epaxial myotome and propose that similar functions may shape the dorsal musculature across vertebrates.

---

## [Decision Letter]

**Decision letter after peer review:**

Thank you for submitting your article "Zic1 advances epaxial myotome morphogenesis to cover the neural tube via Wnt11r" for consideration by *eLife*. Your article has been reviewed by 3 peer reviewers, one of whom is a member of our Board of Reviewing Editors, and the evaluation has been overseen by Marianne Bronner as the Senior Editor. The reviewers have opted to remain anonymous.

Essential revisions:

All three reviewers agree that the study addresses an interesting and underexplored question in developmental biology. The findings are interesting, especially the illustration of the dynamic behaviors of dorsal somitic cells, which form elaborate protrusions, delaminate from their parent somite, and bridge the gap between opposing epaxial myotomes. The reviewers would like the authors to address the minor technical issues outlined in the individual reviews if possible, as well as the following major critiques:

1) The reliance on photomorpholinos to downregulate wnt11r dampens the enthusiasm of the reviewers, as the judgement of its efficacy relies on circumstantial evidence that the phenotype resembles the zic1 phenotype and that exogenously supplied human wnt11 protein partially rescues the phenotype. The reviewers encourage the authors to generate a wnt11r Crispr mutant to validate the morpholino experiments.

2) The identification of a migratory population of dorsal somitic cells is an intriguing observation and the focus of this paper. The fate and function of these cells in epaxial myogenesis, however, is not clear from the current data set and is a gap in the paper. The function of these cells could possibly be explored through an examination of mutants and treatments that prevent their formation or migration.

*Reviewer #1 (Recommendations for the authors):*

Critique:

1) Wnt11r in zebrafish was renamed to wnt11f1 and is the ortholog to human wnt11 (see Postlethwait et al. Evolutionary origin and nomenclature of wnt11- family genes, 2019). The authors should check if medaka wnt11r is the ortholog to human wnt11 and should also be renamed.

2) In zebrafish neither wnt11 or wnt11r mutants die during gastrulation. The authors should generate mutants before concluding that they will not survive past gastrulation, especially since there is no good control for the efficacy of the knockdown of wnt11r by photomorpholinos.

3) Figure 5A: It is unclear to me why zic1:GFP and zic4DSRed fish were generated. Wouldn't the DSRed interfere with the detection of dead PI+ cells during FACS sorting?

4) Figure 5C: Please describe in the result section that dorsal and ventral somites were dissected and that the GFP- cells do not consist of cells from elsewhere in the body.

5) Discussion: there is no discussion on how wnt11r might instruct polarized cell protrusions. Does wnt11r need to be expressed in a gradient? Does it act non-cell autonomously or in a paracrine fashion? Is wnt11r an instructive cue or a permissive cue?

6) Throughout the figures it is hard to see which panels show mutants and which ones show wildtype embryos. Please label better, eg. on the left of the row of images.

*Reviewer #2 (Recommendations for the authors):*

Pigmentation (e.g., Fig1I) makes it difficult to visualize the margins of the epaxial myotomes.

The authors should consider complementing Ph3 labelling with BrdU/Edu labelling for cell birth-dating as a measure of proliferation.

While the medial movement of Zic1:GFP cells to bridge the gap between epaxial myotomes is an interesting observation, the authors do not provide clear evidence of the function of this population in epaxial myotome closure of the neural tube. What is the fate of these cells? Do they contribute to epaxial musculature?

In Figure 4, it is unclear what BFP labeling marks.

The authors provide quantification of the protrusions between WT and Da mutants. The authors should consider a similar quantification for the number of dorsal mesenchymal cells between Da and Wt mutants. Additionally, since the authors have the data already in hand, rendered cross sections might provide another way to clearly visualize differences in the epaxial myotome between WT and mutant fish.

The authors identified Wnt11r as a putative mediator of Zic function during epaxial myotome expansion and hypotheisized Wnt11r functions through the Wnt/ca^2+^ signalling pathway. To test this, they treated embryos with Kn-93. Is it known whether this drug affects off additional pathways? The authors should include a comment on the specificity of the drug.

The authors should further explain the controls used for ChIP-seq experiments.

It would be interesting to know if similar cellular protrusions are present in the ventral margins of hypaxial myotomes, and if the presence of protrusions differ between fin and non-fin levels?

The role of migratory somitic cells to form hypaxial muscles such as those of limb is well described. The identification of migratory somitic cells originating from the Pax3/7+ DM/ECL population in fish is a particularly intriguing observation that seems to raise questions about how programs for migratory cell formation are similar and different between epaxial and hypaxial embryonic contexts.

Line 757 typo "semaphoring" should be "semaphorin".

The author's ChipSeq data sets highlight the position of putative Wnt11r enhancer, and their ATACseq data suggests this region is differentially open in the dorsal somite versus the ventral somite. Given these intriguing data sets, the authors should consider using this region to generate an enhancer-reporter transgenic line to test if this enhancer drives dorsal expression.

*Reviewer #3 (Recommendations for the authors):*

The opening several paragraphs of the discussion include materials that are already covered in the results. It's good to give a synopsis and explain the model, but it would be nice to see this manuscript better connected with the larger literature. Help the readers understand why this work is important. A few potential connections are laid out in the following paragraphs.

I find it striking how similar the cell behaviors are in the epaxial myotome to behaviors shown in the hypaxial myotome of zebrafish. For instance, in both cases researchers have used long-protrusions as evidence of cellular guidance. It would be interesting to see discussion added comparing/contrasting behaviors on the extreme dorsal/ventral ends of differing somites. I will understand if you think this is off-topic, but I think a little discussion about how the epaxial findings related to other muscle groups could give context that broadens the impact of the study.

Even in the narrow literature, some citations are missing. For instance, a quick pubmed of 'zic1 epaxial' gives only three hits; one by the authors and two from another group – the latter two are not cited; these could easily be worked into introduction or discussion.

The paper could also benefit from the authors finding that Da mutants show increased dorsal muscle growth compared to the wild type, even though the mutant muscles do not enclose the dorsal surface. This finding seems like it further supports the author's claim that dorsal closure is not controlled merely by physical constraint; if the authors agree, they could mention this in the discussion.

Presentation issues:

The time-lapse videos could be strengthened by additional annotation of what the reader is supposed to be looking at. The one little arrow in videos 1 and 2 helps some, but it is not enough when there are complex movements amongst a sea of cells and no annotations are provided for videos 3-5, even though arrowheads are mentioned in-text for videos 3 and 4.

I needed to modify videos 3-5 before they played on my new Mac, though they did on a somewhat older Mac. Please double-check the formatting and make sure they are cross compatible with newer systems. The lack of arrowheads also suggests that an incorrect version may have been uploaded.

The animals in Video 3 (especially) and 5 do not hold still, making it difficult to follow what's happening. If feasible amid covid restrictions, it would be nice to see these replaced with time-lapse of immobilized fish.

It is difficult to see protrusions in figure 8b. This panel would benefit from separation of Rhodamine/GFP channels (in addition to the merge) and/or an accompanying illustration.

[Editors' note: further revisions were suggested prior to acceptance, as described below.]

Thank you for resubmitting your work entitled "Zic1 advances epaxial myotome morphogenesis to cover the neural tube via Wnt11" for further consideration by *eLife*. Your revised article has been evaluated by Marianne Bronner (Senior Editor) and a Reviewing Editor.

The manuscript has been significantly improved but there are some remaining issues that need to be addressed, as outlined below:

As noted in the original reviews, the reviewers find the results reported in the manuscript interesting and the data of high quality. The revised manuscript contains new experimental data and the majority of the reviewers' prior concerns are addressed.

The reviewers agree that the requested wnt11 mutant analysis is indeed not informative as the embryos possess convergent extension defects. Even though the wnt11 morpholino experiments possess caveats, their phenotype correlates with the results of several other experiments, suggesting that the morpholinos work as expected. However, the text describing the results should reflect that these are morpholino experiments. Eg., line 337 could be changed to saying that the MO 'supports an important role'.

One major concern with the prior submission was that the fate and function of the dorsal somitic cells bridging opposing myotomes were not described. Here the authors present new data from pharmacological perturbation experiments and targeted laser ablation experiments. These data improve the manuscript and provide evidence that somitic mesenchymal cells have a functional role in myotome extension around the neural tube.

Unfortunately, a more detailed fate analysis of these cells is still missing. A fascinating finding of the paper is the time-lapse analysis of the cells crawling around on top of the neural tube. But it is not entirely clear what those cells become – are they becoming the fin fold or are they becoming muscle? They do not look too different from the fin mesenchyme cells Tom Carney looked at in a paper a few years back, and their appearance does look different from some zebrafish epaxial muscle-forming cells. Since the authors' main conclusion is about muscle, and their main analysis is of cells that we are not sure are becoming muscle, the authors should perform a more conclusive fate analysis.

So far, the fate of somitic mesenchyme cells is only partially addressed through use of mosaic pMTB-membmTagBFP2 expression. The authors extend their previous analysis to argue DM cells contribute to the median fin-fold. Did the authors examine only those somites where mosaic BFP was restricted to the DM and not present in other somitic tissues? This would be important for interpretation of results.

[Editors' note: further revisions were suggested prior to acceptance, as described below.]

Thank you for resubmitting your work entitled "Zic1 advances epaxial myotome morphogenesis to cover the neural tube via Wnt11" for further consideration by *eLife*. Your revised article has been evaluated by Marianne Bronner (Senior Editor) and a Reviewing Editor.

The manuscript has been improved but there are some remaining issues that need to be addressed, as outlined below:

The authors further improved the manuscript by adding fate mapping data of the dorsal DM cells. The fate mapping shows that the dorsally extending cells of interest contribute to the fin-fold mesenchyme and appear to surround blood vessels, however no labeled cells contribute to the myotome itself.

1) These new findings need to be better represented throughout the manuscript. For example, the abstract still reads:

'In wild type, dorsal dermomyotome (DM) cells, progenitors of myotomal cells, reduce their proliferative activity after somitogenesis and subsequently form unique large protrusions extending dorsally, guiding the epaxial myotome dorsally'

Please clarify that dorsal dermomyotome cells are not myotomal progenitors in this context.

2) Given the new fate mapping data, please also make the modified model more explicit throughout the abstract, intro and discussion: that a non-myogenic lineage of pioneer cells non-cell-autonomously guides dorsal myotome growth.

3) Could the authors confirm that the fate mapping experiments consist of 17 individual cells in 17 different embryos?

4) Title: 'Zic1 advances epaxial myotome morphogenesis to cover the neural tube via Wnt11'. Please consider rephrasing the title, which currently does not convey that wnt11 acts on DM cells, which then guide the epaxial myotome. The title suggests that wnt11 acts on epaxial myotome cells directly.

Line 571, 1371: CRISPR-mediated knock-out of wnt11. Please, rephrase knock-out as the analyzed embryos were not stable mutants. F0 CRISPR causes a hodgepodge of mutations and WT cells.

Please soften the language regarding the morpholino experiments as initially requested by the reviewers.

---

## [Author Response]

Essential revisions:All three reviewers agree that the study addresses an interesting and underexplored question in developmental biology. The findings are interesting, especially the illustration of the dynamic behaviors of dorsal somitic cells, which form elaborate protrusions, delaminate from their parent somite, and bridge the gap between opposing epaxial myotomes. The reviewers would like the authors to address the minor technical issues outlined in the individual reviews if possible, as well as the following major critiques:1) The reliance on photomorpholinos to downregulate wnt11r dampens the enthusiasm of the reviewers, as the judgement of its efficacy relies on circumstantial evidence that the phenotype resembles the zic1 phenotype and that exogenously supplied human wnt11 protein partially rescues the phenotype. The reviewers encourage the authors to generate a wnt11r Crispr mutant to validate the morpholino experiments.

Following the reviewers’ suggestion, we designed gRNAs against *wnt11* (we renamed the gene *wnt11* from *wnt11r* after a suggestion by Reviewer 1) and generated *wnt11* crispant embryos (Figure 7 —figure supplement 2A). We found out that development in the crispants were severely affected with a smaller head and truncated body axis at the somitogenesis stage (Figure 7 —figure supplement 2E-F’, arrowheads). This phenotype is known to be secondarily caused by impaired gastrulation movement, and indeed we observed gastrulation defects in the crispants (Figure 7 —figure supplement 2C, D). Because of these defects, we were unable to assess the myotome development in the *wnt11* mutant. However, these phenotypes (impaired gastrulation movement and malformed trunk) were consistent with our observation in the *wnt11* morphants when a high dose of *wnt11* MO was injected, further supporting that the *wnt11* MO inhibits the *wnt11* gene specifically.

In the revised manuscript, we added description on the *wnt11* cripant embryos to justify the MO experiments (Line 305).

“Injection of gRNAs against *wnt11* to knock out the gene revealed that the resultant F0 embryos displayed delayed epiboly movement and subsequent morphological defects, including shorter body axis and impaired trunk development (Figure 7 —figure supplement 2), implying the difficulty of assessment of the *wnt11* function at later stages using the *wnt11* mutant.”

2) The identification of a migratory population of dorsal somitic cells is an intriguing observation and the focus of this paper. The fate and function of these cells in epaxial myogenesis, however, is not clear from the current data set and is a gap in the paper. The function of these cells could possibly be explored through an examination of mutants and treatments that prevent their formation or migration.

We absolutely agree with the suggestion that the function and fate of the dorsal DM and mesenchymal cells are important for the significance of this study. We tested their roles with two additional experiments.

1) At the onset of dorsal somite extension, we inhibited filopodia formation of large protrusions of dorsal DM cells with ML141, a specific inhibitor against Rac1/Cdc42 (Figure 3L, M), and found that the treatment disturbed the dorsal extension of myotomes (Figure 3N). This result suggests that the dorsal DM cells promote the dorsal somite extension by utilizing the large protrusions.

2) We also ablated the mesenchymal DM cells arising at later stages of dorsal somite extension, using a UV laser and revealed that the ablation transiently widened the gap between the left and right myotomes and delayed the dorsal somite extension (Figure 4J-L). We concluded that the mesenchymal DM cells exert a force promoting the advancement of left and right myotomes to cover the neural tube. Thus, DM cells seem to be playing a crucial role throughout dorsal somite extension.

We further investigated the fate of the dorsal DM cells by sparsely labeling those cells with BFP plasmid injection. We found that some cell eventually differentiated into myotome fibers while some became mesenchymal cells in the dorsal finfold (Figure 4 —figure supplement 3C). This result implies that the mesenchymal cells have multiple potentials to become at least myotomal and dermal cells.

We included the description on the function of the dorsal DM and mesenchymal cells in the Results section as follows:

1 (Line 172):

“To investigate the role of large protrusions during the onset of dorsal somite extension, we inhibited filopodia formation using ML141 (Figure 3L, M). ML141 specifically inhibits Cdc42/Rac1 GTPases, which are critical for filopodia formation (Hong et al., 2013; Fantin et al., 2015). Intriguingly, dorsal somites of embryos treated with ML141 extended significantly less dorsally compared to control embryos treated with DMSO (Figure 3N).”

2 (Line 208):

“To examine the function of mesenchymal DM cells, we ablated these cells between the 10th somites of 5.5 dpf Tg(zic1:GFP) embryos with a UV laser (Figure 4J-K’, Figure 4 —figure supplement 3A-B’). Intriguingly, the distance between the left and the right tips of the 10th somites increased after ablating the mesenchymal DM cells while the neighboring 9th somites continued to shorten the gap (Figure 4L). This suggests that mesenchymal DM cells hold the left and the right somite together to promote the dorsal somite extension at this late phase of myotome development. After 16 hours post ablation, dorsal extension of the 10th somites eventually resumed as the mesenchymal DM cells regenerated at the ablation site (Figure 4L).”

Regarding the fate of DM cells, we added a sentence “Interestingly, besides the differentiation into myotome cells, our preliminary data suggest that DM cells also differentiate into mesenchymal fin fold cells, aligning with previous observations in *Xenopus* (Garriock and Krieg, 2007). “ to the Discussion section (Line 423).

Reviewer #1 (Recommendations for the authors):Critique:1) Wnt11r in zebrafish was renamed to wnt11f1 and is the ortholog to human wnt11 (see Postlethwait et al. Evolutionary origin and nomenclature of wnt11- family genes, 2019). The authors should check if medaka wnt11r is the ortholog to human wnt11 and should also be renamed.

We thank the reviewer for the important suggestion. We examined the sequence homology and synteny of medaka *wnt11r* and confirmed that it is the only orthologue of human *WNT11* (there was no *wnt11f1* found in the medaka genome). We rewrote it as *wnt11* throughout the manuscript.

2) In zebrafish neither wnt11 or wnt11r mutants die during gastrulation. The authors should generate mutants before concluding that they will not survive past gastrulation, especially since there is no good control for the efficacy of the knockdown of wnt11r by photomorpholinos.

Following the reviewer’s comment, we generated *wnt11* crispant embryos and found that they have gastrulation defects which lead to impaired trunk development (Figure 7 —figure supplement 2). We also confirmed that the crispant phenotypes were identical to the phenotypes of *wnt11* morphants after injected with a high dose of *wnt11* MO. We thus think that we can evaluate the *wnt11* function using the MO and PhotoMO.

We found that the medaka genome has only one *wnt11* while there was (no *wnt11f1* or a homologue of zebrafish silberblick). We reason that the severe phenotype observed in medaka *wnt11* crispant embryos is probably due to the lack of paralogues (*wnt11f1*).

In the revised manuscript, we added description on the *wnt11* cripant embryos to justify the MO experiments (Line 305).

“Injection of gRNAs against *wnt11* to knock out the gene revealed that the resultant F0 embryos displayed delayed epiboly movement and subsequent morphological defects, including shorter body axis and impaired trunk development (Figure 7 —figure supplement 2), implying the difficulty of assessment of the *wnt11* function at later stages using the *wnt11* mutant.”

3) Figure 5A: It is unclear to me why zic1:GFP and zic4DSRed fish were generated. Wouldn't the DSRed interfere with the detection of dead PI+ cells during FACS sorting?

We apologize for the misleading figure. We used the *zic1:GFP;zic4:DsRed* transgenic line that had been used in the previous chapters. As described in Line 143, DsRed fluorescence in somitic cells is very faint due to the weak activity of the *zic4* promoter. Thus the weak DsRed fluorescence did not interfere with the π staining. To avoid confusion, we removed the picture of BAC construct in Figure 5A.

4) Figure 5C: Please describe in the result section that dorsal and ventral somites were dissected and that the GFP- cells do not consist of cells from elsewhere in the body.

We thank the reviewer for the suggestion. We exclusively isolated somites after removing the rest of the embryos before FACS sorting. Thus GFP+ and – cells in the analyses are all derived from somites. Following the reviewer’s suggestion, we clarified the description on the somite dissection procedure in the Results section (Line 227), methods section (Line 665) and in Figure 5A,B.

5) Discussion: there is no discussion on how wnt11r might instruct polarized cell protrusions. Does wnt11r need to be expressed in a gradient? Does it act non-cell autonomously or in a paracrine fashion? Is wnt11r an instructive cue or a permissive cue?

According to previous studies, the cell autonomy of Wnt11 function is context dependent. In *Xenopus* finfold mesenchymal cells, Wnt11 was suggested to act in a cell-autonomous manner. Meanwhile, other studies have proposed a cell non-autonomous role of Wnt11 in *Xenopus* cranial neural crest migration and chick muscle fiber orientation (De Calisto et al. Development 2005; Gros et al. Nature 2009). In our study, administration of exogenous Wnt11 protein rescued the protrusive behavior of the surrounding DM cells, suggesting that Wnt11 can function cell non-autonomously during dorsal somite extension. Since the dorsal DM cells gradually migrate out of the *wnt11*-positive dorsal somite region during the process, Wnt11 may not work as an attractive (instructive) cue, but likely works as a permissive cue for dorsal migration.

However, our current data do not clearly tell whether Wnt11 functions cell-autonomously or non-cell-autonomously. We thus only added a few sentences to the discussion part Line 457 as follows:

“Whether Wnt11 functions cell-autonomously or not is context dependent (Garriock and Krieg, 2007; De Calisto et al. 2005; Gros et al. 2009). In our study, administration of exogenous Wnt11 protein rescued the protrusive behavior of the surrounding DM cells, suggesting that Wnt11 can function cell non-autonomously during dorsal somite extension.”

6) Throughout the figures it is hard to see which panels show mutants and which ones show wildtype embryos. Please label better, eg. on the left of the row of images.

We appreciate the reviewer’s comment. We added labels on the left of the images in most figures.

Reviewer #2 (Recommendations for the authors):Pigmentation (e.g., Fig1I) makes it difficult to visualize the margins of the epaxial myotomes.

In our previous studies we have used the *Da* mutant strain derived from a natural population which has melanophores. We used the same strain in the current study for consistency of all data. Because the contour of the myotomes was sometimes difficult to see in a single z plane of confocal images as the reviewer pointed out, we carefully examined the z-stack images to determine the precise position of the boundaries. We explain this in the Figure legend, “the contour of the myotomes was drawn based on the Z-stack images of the dorsal myotomes to avoid ambiguity caused by melanophores.“

The authors should consider complementing Ph3 labelling with BrdU/Edu labelling for cell birth-dating as a measure of proliferation.

We tried EdU labelling, but unfortunately it did not work well for medaka embryos under our experimental condition. We thus performed an alternative experiment using immunohistochemistry against another proliferation marker, PCNA, to confirm the results of pH3 staining. In the Results section, we added the sentences accordingly (Line 126):

“Immunohistochemistry against another proliferation marker PCNA confirmed these findings (Figure 2 —figure supplement 1A-C).”

While the medial movement of Zic1:GFP cells to bridge the gap between epaxial myotomes is an interesting observation, the authors do not provide clear evidence of the function of this population in epaxial myotome closure of the neural tube. What is the fate of these cells? Do they contribute to epaxial musculature?

We agree with the reviewer’s suggestion and experimentally investigated the function and fate of the dorsal DM and mesenchymal cells. As described in detail in our response to Essential Revision 2 above, we have changed the manuscript by incorporating the new results (function – Lines 172, 208; fate – Line 423).

In Figure 4, it is unclear what BFP labeling marks.

We have revised Figure 4A-F to indicate more clearly the BFP-labelled somites and dorsally moving cells derived from them.

The authors provide quantification of the protrusions between WT and Da mutants. The authors should consider a similar quantification for the number of dorsal mesenchymal cells between Da and Wt mutants. Additionally, since the authors have the data already in hand, rendered cross sections might provide another way to clearly visualize differences in the epaxial myotome between WT and mutant fish.

We thank the reviewer for the important suggestions. We do agree that the difference between Wt and *Da* mutant mesenchymal DM cells can be clarified by providing cross-sections. In the revised manuscript, we thus added cross-sections to Figure 4A-F.

We also quantified the number of mesenchymal DM cells by analyzing the cross-sections of Tg(*zic1:GFP*) and Tg(*zic1:GFP*);*Da* embryos and incorporated the results into the main text (Figure 4 —figure supplement 1L, left graph). We found that there is a trend towards more mesenchymal cells accumulating between the somites of embryos in Wt background, but the difference was not statistically significant. We thus added a phrase “while the number of mesenchymal cells was not largely affected“ to the Results section (Line 206).

The authors identified Wnt11r as a putative mediator of Zic function during epaxial myotome expansion and hypotheisized Wnt11r functions through the Wnt/ca^2+^ signalling pathway. To test this, they treated embryos with Kn-93. Is it known whether this drug affects off additional pathways? The authors should include a comment on the specificity of the drug.

The specificity of KN-93 inhibition on CaMKII was previously validated (Sumi et al. Biochem Biophys Res Commun. 1991), and the drug has been widely used for inhibition of CaMKII in various studies (Tombes et al., 1995; Wu and Cline, 1998; Garriock and Krieg, 2007; Rothschild et al., 2013). We clarified the specificity of the drug in the main text (Line 355).

The authors should further explain the controls used for ChIP-seq experiments.

We used input chromatin for the control of ChIPmentation. The input chromatin of the ChIP was tagmentated after DNA purification step. This input chromatin was used for calling peaks by MACS2 software. We modified the methods to include the description on the control (Line 745).

It would be interesting to know if similar cellular protrusions are present in the ventral margins of hypaxial myotomes, and if the presence of protrusions differ between fin and non-fin levels?

We thank the reviewer for raising these intriguing questions. For technical reasons, it is hard to observe the extension procedure of the hypaxial myotomes. We speculate that the ventral-most DM cells also exhibit protrusions while they are less active than in dorsal DM cells, since the hypaxial somites show similar characteristics to the epaxial ones in the *Da* mutant embryos. The left and right myotomes cover the neural tube along the entire anterior-posterior axis including the dorsal fin level. It is thus likely that the protrusive activity of the dorsal DM cells can be observed regardless of the presence of fins.

The role of migratory somitic cells to form hypaxial muscles such as those of limb is well described. The identification of migratory somitic cells originating from the Pax3/7+ DM/ECL population in fish is a particularly intriguing observation that seems to raise questions about how programs for migratory cell formation are similar and different between epaxial and hypaxial embryonic contexts.

We agree that the similarity of the migratory behavior in the dorsal and ventral/hypaxial muscle precursor cells derived from DM would be an interesting topic. In the discussion part, we added the following sentences to compare the two systems in detail (Line 408):

“In zebrafish, hypaxial muscle precursors delaminate from the ventral tip of the DM and migrate collectively in a cell stream to the prospective pectoral fin bud region while the cells at the leading edge form long filopodia (Haines et al., 2004; Talbot et al., 2019). Thus, DM cells, regardless of whether dorsal or ventral, seem to have the potential to form protrusions and to exhibit migratory behavior. However, regulation of migratory behavior could be different as the migration of hypaxial muscle precursors is mediated by the receptor tyrosine kinase Met (Haines et al., 2004) which is not expressed in zebrafish dorsal DM cells.”

Line 757 typo "semaphoring" should be "semaphorin".

We apologize for the typo. We have corrected the spelling of semaphorin.

The author's ChipSeq data sets highlight the position of putative Wnt11r enhancer, and their ATACseq data suggests this region is differentially open in the dorsal somite versus the ventral somite. Given these intriguing data sets, the authors should consider using this region to generate an enhancer-reporter transgenic line to test if this enhancer drives dorsal expression.

We appreciate the reviewer’s helpful suggestion. To prove that the region acts as a *wnt11* enhancer, it will be necessary to establish the enhancer-reporter line. However, we think that the enhancer analysis of the *wnt11* gene is beyond the scope of this study. We therefore did not perform experiments to address this issue. In this regard, the use of the word “enhancer” may not be appropriate, and we therefore rewrote the Results section as follows (Line 291):

“These sites were more accessible in dorsal somites than in ventral somites, suggesting that Zic1 regulates *wnt11* directly (Figure 6C).”

Reviewer #3 (Recommendations for the authors):The opening several paragraphs of the discussion include materials that are already covered in the results. It's good to give a synopsis and explain the model, but it would be nice to see this manuscript better connected with the larger literature. Help the readers understand why this work is important. A few potential connections are laid out in the following paragraphs.

We thank the reviewer for the valuable advice. We shortened the first paragraphs in the discussion and added the following sentence to clarify the significance of our work (Line 388).

“To our knowledge, our work demonstrates for the first time that neural tube coverage by myotomes is driven by active cell movement, rather than being passively achieved by muscle growth.”

I find it striking how similar the cell behaviors are in the epaxial myotome to behaviors shown in the hypaxial myotome of zebrafish. For instance, in both cases researchers have used long-protrusions as evidence of cellular guidance. It would be interesting to see discussion added comparing/contrasting behaviors on the extreme dorsal/ventral ends of differing somites. I will understand if you think this is off-topic, but I think a little discussion about how the epaxial findings related to other muscle groups could give context that broadens the impact of the study.

We appreciate the reviewer’s suggestion. This interesting point was also raised by Reviewer 2, and we have incorporated the comparison between the dorsal and the ventral/hypaxial muscle precursor cells into the Discussion section as follows (Line 408):

“In zebrafish, hypaxial muscle precursors delaminate from the ventral tip of DM and migrate to the prospective pectoral fin bud region, extending filopodial protrusions (Haines et al. 2004; Talbot et al. 2019). DM cells, regardless of whether dorsal or ventral, could thus generally have potential to exhibit migratory behavior with active protrusions. However, regulation of migratory behavior could be different as the migration of hypaxial muscle precursors is mediated by the receptor tyrosine kinase Met (Haines et al. 2004) while it is not expressed in zebrafish dorsal DM cells. Furthermore, hypaxial muscle precursors form tightly packed cell streams migrating out from the somite, whereas dorsal DM cells migrate individually.”

Even in the narrow literature, some citations are missing. For instance, a quick pubmed of 'zic1 epaxial' gives only three hits; one by the authors and two from another group – the latter two are not cited; these could easily be worked into introduction or discussion.

We are sorry for the insufficient citations. We added several more papers referencing the functions of *zic1/zic4* during somite development and specifically during myotome development in fish (Lines 40, 49).

The paper could also benefit from the authors finding that Da mutants show increased dorsal muscle growth compared to the wild type, even though the mutant muscles do not enclose the dorsal surface. This finding seems like it further supports the author's claim that dorsal closure is not controlled merely by physical constraint; if the authors agree, they could mention this in the discussion.

We fully agree with the reviewer’s opinion and incorporated this discussion in the Results section to justify our analyses of cellular behavior, as well as in the Discussion section.

(Line 136)

“In the *Da* mutant, the myotome is unable to cover the neural tube despite increased dorsal myotome growth at the hatching stage (Figure 1L, Figure 2). This suggest that additionally to physical extension an active process might support the dorsal movement of somites.”

(Line 391):

“Consistent with this, in the absence of Zic1 activity, the myotome is unable to cover the neural tube at the hatching stage despite increased dorsal myotome growth (Figure 1L, Figure 2), supporting the involvement of the active cellular mechanism in this process.”

Presentation issues:The time-lapse videos could be strengthened by additional annotation of what the reader is supposed to be looking at. The one little arrow in videos 1 and 2 helps some, but it is not enough when there are complex movements amongst a sea of cells and no annotations are provided for videos 3-5, even though arrowheads are mentioned in-text for videos 3 and 4.

We apologize for the confusion. The legends of videos 3 and 4 in the result part do belong to the videos 1 and 2. There must have happened a mix-up of the video files. We reviewed the videos and legends in the main text again and uploaded the videos correctly.

I needed to modify videos 3-5 before they played on my new Mac, though they did on a somewhat older Mac. Please double-check the formatting and make sure they are cross compatible with newer systems. The lack of arrowheads also suggests that an incorrect version may have been uploaded.

We thank the reviewer for pointing out this issue. We modified the format of the videos and verified that they can be now played on new and old Mac computers, as well as on Windows computers. The lack of arrowheads in the videos 3 and 4 was due to the mishandling of annotations of the files (as explained in the comment above) and has been verified too.

The animals in Video 3 (especially) and 5 do not hold still, making it difficult to follow what's happening. If feasible amid covid restrictions, it would be nice to see these replaced with time-lapse of immobilized fish.

We appreciate the reviewer’s comment. We tried to fix the position of the embryos but we failed to do so completely for the following technical reasons. The movement of the medaka embryos in the videos consists of three factors: muscle contraction, yolk contraction and body axis growth. We successfully inhibited the muscle movement by injecting *bungarotoxin* mRNA which blocks nicotinic acetylcholine receptors (Swinburne et al. PLoS One 2015; Lischik et al. PLoS One 2019). However, the yolk contraction movement, which is unique to the early stages (from epiboly to mid-somitogenesis stages) of medaka embryos and is not observed in zebrafish, was difficult to be completely inhibited. The slight twitching of the embryos in the movies are due to this yolk movement. Furthermore, the movement to the left in videos 3 and 4 is due to the rapid body axis growth of the embryos during that stage, although we minimized the growth effect by registering the images by ImageJ.

It is difficult to see protrusions in figure 8b. This panel would benefit from separation of Rhodamine/GFP channels (in addition to the merge) and/or an accompanying illustration.

We fully agree with the reviewer and separated the channels in Figure 8B.

We also reorganized the formality of every figure legend to enhance the readability of them in the revised manuscript but did not highlight the changes using the Word proofreading function. Other modifications are highlighted.

[Editors' note: further revisions were suggested prior to acceptance, as described below.]

The manuscript has been significantly improved but there are some remaining issues that need to be addressed, as outlined below:As noted in the original reviews, the reviewers find the results reported in the manuscript interesting and the data of high quality. The revised manuscript contains new experimental data and the majority of the reviewers' prior concerns are addressed.The reviewers agree that the requested wnt11 mutant analysis is indeed not informative as the embryos possess convergent extension defects. Even though the wnt11 morpholino experiments possess caveats, their phenotype correlates with the results of several other experiments, suggesting that the morpholinos work as expected. However, the text describing the results should reflect that these are morpholino experiments. Eg., line 337 could be changed to saying that the MO 'supports an important role'.One major concern with the prior submission was that the fate and function of the dorsal somitic cells bridging opposing myotomes were not described. Here the authors present new data from pharmacological perturbation experiments and targeted laser ablation experiments. These data improve the manuscript and provide evidence that somitic mesenchymal cells have a functional role in myotome extension around the neural tube.Unfortunately, a more detailed fate analysis of these cells is still missing. A fascinating finding of the paper is the time-lapse analysis of the cells crawling around on top of the neural tube. But it is not entirely clear what those cells become – are they becoming the fin fold or are they becoming muscle? They do not look too different from the fin mesenchyme cells Tom Carney looked at in a paper a few years back, and their appearance does look different from some zebrafish epaxial muscle-forming cells. Since the authors' main conclusion is about muscle, and their main analysis is of cells that we are not sure are becoming muscle, the authors should perform a more conclusive fate analysis.So far, the fate of somitic mesenchyme cells is only partially addressed through use of mosaic pMTB-membmTagBFP2 expression. The authors extend their previous analysis to argue DM cells contribute to the median fin-fold. Did the authors examine only those somites where mosaic BFP was restricted to the DM and not present in other somitic tissues? This would be important for interpretation of results.

We agree with the reviewers and indeed could not exclude the possibility that other somitic cells besides the dorsal DM cells were also labeled during our lineage tracing experiment using the BFP plasmid. To tackle this issue, we set up another experiment utilizing KikGR (photoconvertible protein)-mediated photoconversion, which enables spatiotemporally controlled labeling of dorsal DM cells at a single-cell level. We injected *kikGR* mRNA into 1-cell stage embryos and photoconverted single dorsal DM cells specifically with a UV laser. Specific labeling of the dorsal DM cells was confirmed by visual inspection of z-stack images using confocal microscopy (Figure 4M). We tracked their fate until the hatching stage (9 days postfertilization).

We found that the labeled cells contributed to blood vessels (presumably mural cells surrounding endothelial cells; Figure 4N, N’) and dorsal finfold mesenchyme (Figure 4O, O’), which is in accordance with previous studies revealing their somitic origins (Ando et al., Development 2016; Lee et al., Development 2013; Garriock and Krieg, 2007). However, we did not observe any labeled myotomal cells. These results suggest that these dorsal DM cells, guiding the myotome to cover the neural tube, do not contribute to the myotome themselves, at least at the hatching stage.

In the manuscript, we replaced the previous figures for fate mapping with these new figures (Figures 4M-O’) and described them in the results and Discussion sections as follows:

(Line 220, Results):

“Finally, we examined the fate of the dorsal DM cells that derive the mesenchymal cells. We employed a photoconversion technique mediated by a photoconvertible protein KikGR to specifically label a single dorsal DM cell exhibiting protrusions at the tip of a 10^th^ or 20^th^ somite during dorsal somite extension (Figure 4M, arrowhead) and tracked them until the hatching stage. We found that the labeled dorsal DM cells eventually differentiated into cells surrounding blood vessels (likely mural cells, n = 12/17 for 10^th^ and 20^th^ somites; Figure 4N, N’, arrowheads) and mesenchymal cells in the dorsal fin fold (n = 4/8 for 20^th^ somites; Figure 4O, O’, arrowheads); however, we did not observe any labeled axial muscles (n = 0/17 for 10^th^ and 20^th^ somites). This suggests that the dorsal DM cells at the tip of somites, guiding myotome extension, do not become myotomal cells themselves at least until the end of embryonic development (hatching stage, 9 dpf).”

(Line 420, Discussion):

“Interestingly, our lineage tracing experiment revealed that the dorsal DM cells at the tip of somites differentiated into cells of non-myotomal lineages including mural cells of blood vessels and mesenchymal cells in the dorsal fin fold. These observations are consistent with previous studies reporting their somitic origins in zebrafish and *Xenopus* (Garriock and Krieg, 2007; Lee et al., 2013; Ando et al., 2016). However, our data also showed that these DM cells do not appear to contribute to the myotome at the hatching stage. We thus propose that the neural tube coverage is driven by a specific population of dorsal DM cells which do not contribute to myotomal cells during embryonic development.”

[Editors' note: further revisions were suggested prior to acceptance, as described below.]

The manuscript has been improved but there are some remaining issues that need to be addressed, as outlined below:The authors further improved the manuscript by adding fate mapping data of the dorsal DM cells. The fate mapping shows that the dorsally extending cells of interest contribute to the fin-fold mesenchyme and appear to surround blood vessels, however no labeled cells contribute to the myotome itself.1) These new findings need to be better represented throughout the manuscript. For example, the abstract still reads:'In wild type, dorsal dermomyotome (DM) cells, progenitors of myotomal cells, reduce their proliferative activity after somitogenesis and subsequently form unique large protrusions extending dorsally, guiding the epaxial myotome dorsally'Please clarify that dorsal dermomyotome cells are not myotomal progenitors in this context.

We appreciate the reviewers’ suggestion and have modified the text to underline the difference between the commonly recognized role of DM cells as myotomal progenitors and our novel finding of the DM cells as a non-myogenic subpopulation as follows:

(Line 17, abstract): “In wild type, dorsal dermomyotome (DM) cells reduce their proliferative activity after somitogenesis. Subsequently, a subset of DM cells, which does not differentiate into the myotome population, begins to form unique large protrusions extending dorsally to guide the epaxial myotome dorsally.”

(Line 44, introduction): “While we have a detailed understanding of how the myotome, precursors of epaxial and hypaxial muscles, differentiates from a somitic compartment called the dermomyotome (DM)…”

(Line 83, introduction): “We also found that these DM cells form an subpopulation that gives rise to non-myotomal cell lineages during embryonic development.”

(Line 392, discussion): “DM cells have been known to serve as a progenitor pool for myotomal and dermal cells (Ben-Yair and Kalcheim, 2005; Hollway *et al.*, 2007; Stellabotte and Devoto, 2007). In our study, we showed that DM cells at the tip of the dorsal somite form unique large protrusions, guiding the myotome towards the top of the neural tube. Furthermore, these DM cells do not give rise to myotomal cells during embryonic development.”

2) Given the new fate mapping data, please also make the modified model more explicit throughout the abstract, intro and discussion: that a non-myogenic lineage of pioneer cells non-cell-autonomously guides dorsal myotome growth.

Following the reviewers’ comment, we have highlighted the modified model involving the non-myogenic population of DM cells as shown below:

(Line 26, abstract): “We propose that dorsal extension of the epaxial myotome is guided by a non-myogenic subpopulation of DM cells and that *wnt11* empowers the DM cells to drive the coverage of the neural tube by the epaxial myotome.”

(Line 89, introduction): “We thus propose an unprecedented process of epaxial myotome morphogenesis driven by a non-myogenic population of DM cells during embryogenesis.”

(Line 398, discussion): “We thus revealed a novel role of non-myogenic DM cells during epaxial myotome morphogenesis.”

3) Could the authors confirm that the fate mapping experiments consist of 17 individual cells in 17 different embryos?

In the fate mapping experiments, we mostly labeled individual DM cells in two different positions (10^th^ and 20^th^ somites) of single embryos and examined the effect of somite position on the myotomal contribution of DM cells. We labeled individual cells in 8 embryos at both 10^th^ and 20^th^ somites, and in 1 embryo we labeled an individual cell only at the 10^th^ somite. During the experiment the labeled cells stayed in the same region during their differentiation, and we never observed movement of the labeled cells over a long distance to intermingle with other labeled cells from a different position.

To clarify this point, we modified the description on the sample size in the Results section (Lines 235, 238), as well as added a sentence in the Materials and methods (Line 556).

4) Title: 'Zic1 advances epaxial myotome morphogenesis to cover the neural tube via Wnt11'. Please consider rephrasing the title, which currently does not convey that wnt11 acts on DM cells, which then guide the epaxial myotome. The title suggests that wnt11 acts on epaxial myotome cells directly.

We have now changed the title to “Wnt11 acts on dermomyotome cells to guide epaxial myotome morphogenesis” to represent the role of Wnt11 more clearly.

We have also rewritten the impact statement to indicate the significance of our findings more explicitly.

“A medaka mutant revealed that Wnt11 promotes formation of uniquely large protrusions from non-myogenic dorsal dermomyotome cells, which guide the epaxial myotome dorsally to achieve the coverage of the neural tube.”

Line 571, 1371: CRISPR-mediated knock-out of wnt11. Please, rephrase knock-out as the analyzed embryos were not stable mutants. F0 CRISPR causes a hodgepodge of mutations and WT cells.

We fully agree with the reviewers’ comment and have rephrased the description as follows:

(Line 586, Materials and methods subtitle): “*wnt11* sgRNA injection”

(Line 1373, Figure 7 —figure supplement 2 title): “Injection of *wnt11* sgRNAs results in delayed epiboly movement, and impaired body axis and trunk development.”

Furthermore, we also added a few words to avoid confusion in the main text:

(Line 326): “Injection of single guide RNAs (sgRNAs) against *wnt11* to knock out the gene revealed that the resultant genetically mosaic F0 embryos displayed delayed epiboly movement and subsequent morphological defects…”

Please soften the language regarding the morpholino experiments as initially requested by the reviewers.

Following the reviewers’ suggestion, we have now softened the expression of sentences claiming the function of *wnt11* based on the morpholino experiments as below:

(Line 356): “Overall, knock-down of the Zic1 target gene *wnt11* recapitulated the phenotype of *Da* DM cells (Figure 3G-G’’), suggesting the crucial role of Wnt11 in regulating protrusion formation of DM cells.”

(Line 451): “Our data suggest that in medaka dorsal somites Wnt11 exerts its effect through promotion of protrusion formation and down-regulation of cell proliferation in the dorsal DM.”

(Line 480): “Wnt11 may not be a sole factor for dorsal somite extension, although it was suggested to be crucial in the present study.”